# URBANGRAPH: PHYSICS-INFORMED SPATIO-TEMPORAL DYNAMIC HETEROGENEOUS GRAPHS FOR URBAN MICROCLIMATE PREDICTION

**Weilin Xin**[1,†]  **Chenyu Huang**[2,†]  **Peilin Li**[1]  **Jing Zhong**[3]  **Jiawei Yao**[2,*]

[1]National University of Singapore  [2]Tongji University  [3]Tsinghua University

## ABSTRACT

With rapid urbanization, predicting urban microclimates has become critical, as it affects building energy demand and public health risks. However, existing generative and homogeneous graph approaches fall short in capturing physical consistency, spatial dependencies, and temporal variability. To address this, we introduce UrbanGraph, a framework founded on a novel structure-based inductive bias. Unlike implicit graph learning, UrbanGraph transforms physical first principles into a dynamic causal topology, explicitly encoding time-varying causalities (e.g., shading and convection) directly into the graph structure to ensure physical consistency and data efficiency. Results show that UrbanGraph achieves state-of-the-art performance across all baselines. Specifically, the use of explicit causal pruning significantly reduces the model's floating-point operations (FLOPs) by 73.8% and increases training speed by 21% compared to implicit graphs. Our contribution includes the first high-resolution benchmark for spatio-temporal microclimate modeling, and a generalizable explicit topological encoding paradigm applicable to urban spatio-temporal dynamics governed by known physical equations.

## 1 INTRODUCTION

Urban microclimate prediction is crucial for urban sustainability and public health (Grant et al., 2025; He et al., 2024). This task represents a broad class of spatio-temporal urban physical field prediction problems, such as urban wind field simulation and pollutant dispersion forecasting. The core challenge of these problems is that the physical state at any point in urban space is determined by the collective interactions among numerous and diverse urban entities (e.g., buildings, vegetation) through time-varying physical processes such as radiation and convection (Coutts et al., 2013; de Abreu-Harbich et al., 2015; Irmak et al., 2017; Abd Elraouf et al., 2022). While high-fidelity physics-based numerical simulations, such as Computational Fluid Dynamics (CFD), are the standard approach for solving such problems, their immense computational overhead makes them infeasible for large-scale, time-series prediction tasks. Therefore, exploring computationally efficient data-driven methods to strike a balance between prediction accuracy and efficiency has become an essential research direction.

Although data-driven methods are promising, they still face challenges in accurately modeling the underlying physical processes. While urban data is spatially discretized, physical interactions are often non-local (e.g., shadows skipping intermediate spaces) and anisotropic (e.g., directional wind flow). Grid-based models (CNNs) struggle to capture these long-range, irregular dependencies without excessive depth.(Carter et al., 2016; Kemppinen et al., 2024). Graph Neural Networks (GNNs) offer a more natural framework for modeling the spatial dependencies among urban entities. However, existing GNN-based approaches often lack physical consistency. They typically employ a uniform message-passing mechanism that cannot distinguish between different physical processes, such as vegetation evapotranspiration and building shading (Zhao et al., 2021). Furthermore, these methods struggle to model temporal variability. They mostly rely on a fixed graph structure, which is incapable of representing how physical processes evolve in real-time in response to changing environmental conditions. Consequently, there is a pressing need in the field for a unified framework capable of explicitly modeling multiple physical processes and their temporal evolution.

---

[*]Corresponding author. [†] Equal contribution.

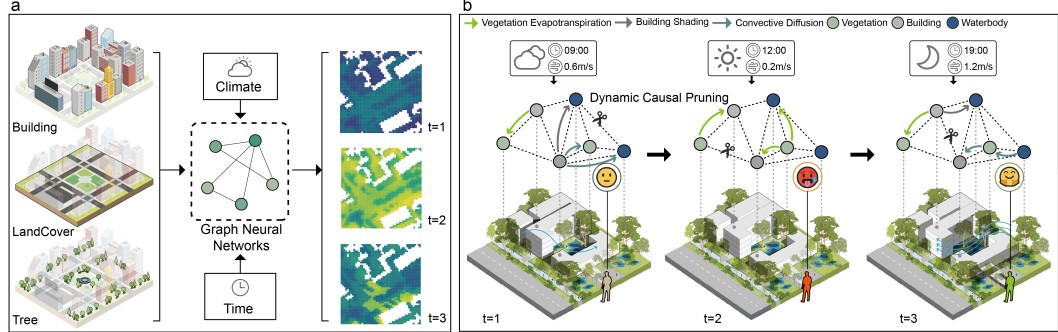

Figure 1: Overview of the UrbanGraph framework. (a) Spatio-Temporal Pipeline: The model transforms rasterized urban features and dynamic weather conditions into high-resolution microclimate heatmaps via GNN learning. (b) Physics-Informed Topology Construction: Physical laws (e.g., shading, diffusion) serve as hard structural constraints to dynamically prune non-causal edges from the candidate set, constructing a sparse and physically consistent graph topology.

Addressing this gap necessitates a fundamental shift in how we model urban dynamics. The core challenge lies in designing a structure-based inductive bias capable of encoding multiple, independent, and time-varying physical processes. (i) Standard graph topologies struggle to abstract continuous physical fields (e.g., radiative transfer and fluid dynamics) into a discrete representation without losing critical causal information. (ii) Furthermore, processing such complex graph sequences requires a neural architecture that can disentangle the diverse physical interactions. The model must not only handle the dual dynamics of both node features and graph topology but also differentiate between distinct physical operators (e.g., shading vs. convection), striking a balance between physical interpretability and computational efficiency(Heo et al.).

To address these challenges, we propose UrbanGraph (see Figure 1), a framework founded on a novel structure-based inductive bias for spatio-temporal modeling. Unlike standard approaches that rely on implicit latent graph learning, UrbanGraph transforms physical first principles into a dynamic causal topology. By explicitly encoding time-varying causalities—such as solar shading and anisotropic wind convection—directly into the graph structure, we impose a hard structural constraint that forces the model's receptive field to align with the actual physical domain of influence. This explicit causal encoding effectively reduces the hypothesis space, preventing the model from learning spurious correlations from noise. Subsequently, we design a spatio-temporal graph network where the heterogeneous message-passing mechanism functions as a physical operator approximator, assigning dedicated learnable parameters to disentangle distinct physical processes(Schlichtkrull et al., 2017).

To summarize, we make the following contributions:

- We propose a structure-based inductive bias via dynamic topological reconfiguration. By explicitly encoding time-varying physical processes into the topology of a dynamic heterogeneous graph, this method offers a novel pathway for injecting time-evolving causal knowledge into graph learning as a hard constraint.

- We develop a dynamic heterogeneous graph neural network architecture that efficiently learns from complex graph sequences. Through a specialized heterogeneous message-passing mechanism, the model achieves physical operator decoupling, allowing for precise modeling of distinct environmental interactions.Comprehensive experiments demonstrate that the architecture achieves state-of-the-art performance in both prediction accuracy and computational efficiency. Compared to four categories of baselines, it improves accuracy by up to 10.8% (in R²) and efficiency by 17.0% (in FLOPs).

- We provide a new, well-validated perspective for modeling urban systems. Results quantitatively demonstrate that the heterogeneous and dynamic mechanisms are key to the performance improvement, contributing gains of 3.5% and 7.1%, respectively. Furthermore, the architecture's results on multiple target variables show strong generalization capabilities.

## 2 RELATED WORK

**Classical Microclimate Prediction Methods**. Classical methods for urban microclimate prediction can be broadly categorized into two types. The first consists of physics-based simulation models, such as CFD and ENVI-met (Toparlar et al., 2017; Tsoka et al., 2018; Liu et al., 2021; Barros Moreira de Carvalho & Bueno da Silva, 2024). These methods offer high physical fidelity but suffer from immense computational overhead, making them impractical for large-scale, long-term time-series prediction tasks. The second category comprises data-driven approaches, including traditional machine learning (Arulmozhi et al., 2021; Alaoui et al., 2023) and grid-based deep learning models like CNNs (Kumar et al., 2021; Kastner & Dogan, 2023; Fujiwara et al., 2024). While these methods are computationally efficient, the former struggle to capture complex spatial dependencies, and the latter are constrained by the Euclidean data assumption, making them unable to process the inherently non-structural geometry of urban environments.While generic GNNs (Kipf & Welling, 2016; Xu et al., 2019; Zhou et al., 2020) offer a framework for spatial dependencies, they lack the inductive bias to abstract continuous fields. UrbanGraph addresses this by explicitly bridging the gap between continuous field dynamics and discrete graph topology.

**Physics-Informed Methods**. The physics-informed approach incorporates fundamental knowledge into the learning process (Karniadakis et al., 2021). This integration is achieved primarily through three bias types. The *learning bias* uses soft constraints (e.g., PDE residuals in the loss function) but incurs significant training overhead (Shao et al., 2023; Taghizadeh et al., 2025). The *observational bias* leverages physics to modify input features (e.g., by extracting high-level physical indicators) (Pan et al., 2025). A third type is the *inductive bias*, which imposes hard constraints via network modules or graph structures to simulate physical processes (Xue; Qu et al., 2023; Gao et al., 2024). However, this traditional hard-constraint approach often sacrifices model flexibility. UrbanGraph addresses this challenge by adopting a novel, efficient form of the inductive bias. By imposing physics as a dynamic structural constraint via graph topology, UrbanGraph ensures consistency and interpretability without sacrificing necessary flexibility or incurring the PDE solver overhead associated with the learning bias.

**Heterogeneous Graph Methods**. In the context of urban physical field prediction, Graph Neural Networks typically simplify the complex urban system into a homogeneous graph (Yu et al., 2024; Zheng & Lu, 2024). However, this simplification limits the model's fidelity and interpretability. Heterogeneous graphs, which consist of multiple types of nodes and edges, can represent the rich semantic relationships in complex systems (Schlichtkrull et al., 2017; Zhang et al., 2019; Zhao et al., 2021). By designing type-aware message-passing mechanisms, Heterogeneous Graph Neural Networks (HGNNs) have achieved success in various tasks, such as quantifying road network homogeneity (Xue et al., 2022), perceiving urban spatial heterogeneity (Xiao et al., 2023), learning urban region representations (Kim & Yoon, 2025), predicting the interactive behaviors of traffic participants (Li et al., 2021), and uncovering the dynamics of building carbon emissions (Yap et al., 2025). We bridge this gap by leveraging heterogeneity for physical operator decoupling, assigning distinct learnable operators to fundamentally different environmental interactions.

**Dynamic Graph Methods**. In applications for urban physical field prediction, Graph Neural Networks often rely on a static graph topology to represent the spatial relationships between entities (Mandal & Thakur, 2023; Shao et al., 2024; Xu et al., 2024). However, this assumption conflicts with physical reality, as the scope and intensity of physical processes (e.g., building shading) are determined in real-time by external environmental factors (e.g., solar position). Dynamic Graph Neural Networks (DGNNs) provide a more realistic framework for this problem (Skarding et al., 2020; Zheng et al., 2024). DGNNs have become a mainstream and effective approach for handling other urban tasks with time-varying interactions, particularly in traffic forecasting, demonstrating their potential in the field of urban computing (Zhao et al., 2020; Xie et al., 2020; Bui et al., 2022). In these mainstream applications, the evolution of the graph is typically treated as a data-driven, observational phenomenon (Li et al., 2019b; Jin et al., 2020). In physical field prediction tasks, however, the graph topology (e.g., shading relationships) is explicitly reconfigured at each timestep by exogenous physical first principles. Consequently, we propose a First-Principle-Driven approach that utilizes dynamic topological reconfiguration as a causal pruning mechanism to explicitly model time-varying interactions.

## 3 PRELIMINARY

**Target Variables**. The Universal Thermal Climate Index (UTCI) (Jendritzky et al., 2012) and the Physiological Equivalent Temperature (PET) (Matzarakis et al., 1999) represent the isothermal air temperature that would elicit the same physiological stress response. Air Temperature (AT) is the most direct measure of atmospheric heat. Mean Radiant Temperature (MRT) quantifies the radiative heat exchange between the human body and its surrounding surfaces, such as sunlit pavements or shaded building facades. Wind Speed (WS) primarily affects convective heat loss and the efficiency of evaporative cooling from the skin surface. Relative Humidity (RH) determines the efficiency of the body's primary cooling mechanism: sweat evaporation.

**ENVI-met model**. The data in this paper were generated using the ENVI-met model. ENVI-met is a high-resolution, three-dimensional, non-hydrostatic numerical model widely recognized for simulating surface-plant-air interactions within complex urban structures. The model captures the feedback mechanisms among different urban elements by coupling an atmospheric model with detailed soil and vegetation models. This enables it to accurately simulate how solid boundaries ('hard' boundaries), such as building walls, and porous obstructions ('soft' boundaries), such as vegetation canopies, alter local airflow, temperature, and humidity. The fundamental equations governing these processes are detailed in Appendix A.

**Problem Formulation**. We model the urban environment by discretizing Geographic Information System (GIS) data into grid cells, where each cell is treated as a node $v \in \mathbb{V}$. The state of the environment is represented by a sequence of dynamic heterogeneous graphs $\{\mathcal{G}_t\}$, where the graph at timestep t is defined as $\mathcal{G}_t = (\mathbb{V}, \mathcal{E}_t, \mathbb{R})$. Here, $\mathbb{V}$ is the static set of nodes, $\mathbb{R}$ is the static set of relation types (e.g., 'covered by shadow from cell'), and $\mathcal{E}_t$ is the set of edges that varies with time. The static node feature matrix $\boldsymbol{X} \in \mathbb{R}^{|\mathbb{V}| \times 8}$ explicitly encodes pixel-wise GIS properties: Building Height, Tree Height and Land Cover Type. Additionally, $\boldsymbol{u}_t$ represents the dynamic global context, comprising meteorological forcing data: Solar Radiation, Solar Position, Ambient Temperature, Humidity, Wind Speed, and Wind Direction.

For any one of the six target variables, denoted by k, given a sequence of historical graph observations of length $T_{hist}$, $\{\mathcal{G}_t\}_{t=t_0-T_{hist}+1}^{t_0}$, and the corresponding sequence of context vectors $\{\boldsymbol{u}_t\}_{t=t_0-T_{hist}+1}^{t_0}$, the model aims to learn a specialized mapping function $\mathcal{F}^{(k)}(\cdot)$ to predict the values of this specific variable for the next $T_{pred}$ timesteps:

$$\left\{ \hat{y}_{t_0+1}^{(k)}, \ldots, \hat{y}_{t_0+T_{\text{pred}}}^{(k)} \right\} = \mathcal{F}^{(k)} \left( \{\mathcal{G}_t\}_{t=t_0-T_{\text{hist}}+1}^{t_0}, \{u_t\}_{t=t_0-T_{\text{hist}}+1}^{t_0}, X \right) \tag{1}$$

where $\hat{\boldsymbol{y}}_\tau^{(k)} \in \mathbb{R}^{|\mathbb{V}|}$ is the predicted vector for the target variable k at a future timestep $\tau$.

**Relational graph convolutional networks**. RGCNs are an extension of GCNs, initially developed for tasks such as link prediction and entity classification. They are specifically designed to handle multi-relational graph data. The core idea is to learn distinct feature transformations for different types of relationships between nodes. The forward-pass update of a single RGCN layer is defined as:

$$\boldsymbol{h}_i^{(l+1)} = \sigma \left( \sum_{r \in \mathbb{R}} \sum_{j \in \mathbb{N}_i^r} \frac{1}{c_{i,r}} \boldsymbol{W}_r^{(l)} \boldsymbol{h}_j^{(l)} + \boldsymbol{W}_0^{(l)} \boldsymbol{h}_i^{(l)} \right) \tag{2}$$

where $\boldsymbol{h}_i^{(l)} \in \mathbb{R}^{d^{(l)}}$ is the hidden state of node $v_i$ in the $l$-th layer, and $d^{(l)}$ is the dimensionality of the representation at this layer. $\mathbb{N}_i^r$ denotes the set of neighbors of node $v_i$ under relation $r \in \mathbb{R}$. $\boldsymbol{W}_r^{(l)}$ is a learnable, relation-specific weight matrix that allows the model to distinguish between different types of relations, and $\boldsymbol{W}_0^{(l)}$ is the weight matrix for the self-connection. $\sigma$ represents an element-wise activation function (e.g., PReLU), and $c_{i,r}$ is a problem-specific normalization constant that can either be learned or preset (e.g., $c_{i,r} = \mathbb{N}_i^r$).

## 4 METHOD

Our proposed UrbanGraph framework consists of two core components: a physics-informed graph representation and a spatio-temporal dynamic relational graph network. To rigorously evaluate the

effectiveness of our approach, we first generated a large-scale spatio-temporal dataset through high-fidelity physical simulations, the detailed generation process and parameter configurations of which are described in Appendix B.1. Second, to address the challenge that urban systems exhibit high heterogeneity in both spatial and temporal dimensions, we detail our physics-informed graph representation in Section 4.1, which is designed to efficiently capture the underlying physical interactions among different urban elements. Finally, in Section 4.2, we introduce the UrbanGraph architecture, which explicitly leverages the time-varying relationships between urban elements to perform node prediction tasks.

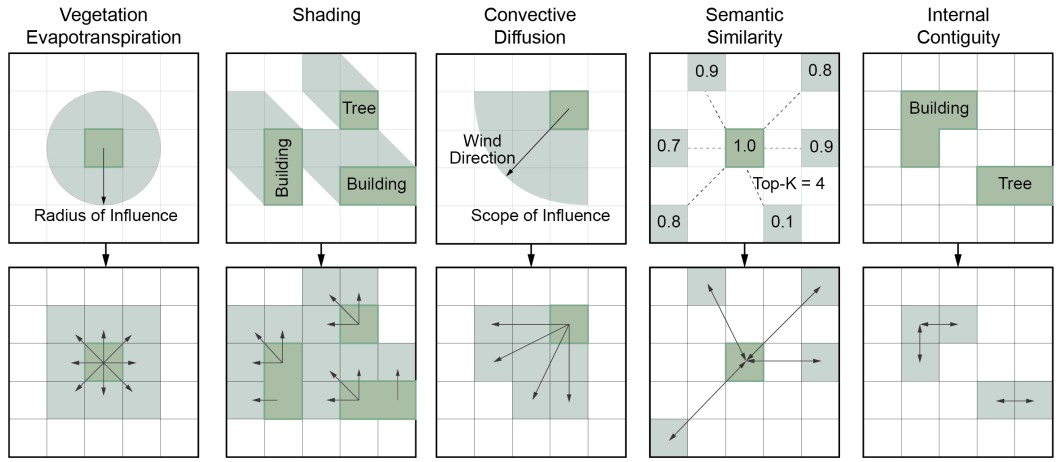

Figure 2: An illustrative overview of the five edge types used in our graph representation. Dynamic edges are derived from physical processes like shadowing and wind, while static edges are based on spatial proximity, feature similarity, and object integrity.

### 4.1 PHYSICS-INFORMED GRAPH REPRESENTATION

For the graph at any given timestep t, $\mathcal{G}_t = (\mathbb{V}, \mathcal{E}_t, \mathbb{R})$, its edge set $\mathcal{E}_t$ is reconstructed based on the environmental conditions of the current hour. This process is designed to explicitly capture the physical mechanisms that govern the spatial distribution of microclimate factors. The edge set $\mathcal{E}_t$ encodes five distinct types of relationships, which are categorized into two main classes: static and dynamic. Figure 2 provides a visual illustration of the construction mechanisms for these five edge types.

**Physics-Informed Dynamic Edges**. To explicitly model time-varying physical processes, we introduce three types of dynamic edges whose connections are updated hourly:

SHADING. This edge type encodes directional radiative obstruction. By establishing links based on geometric line-of-sight, we strictly enforce the dependency of the shadowed region on the occluding object. A directed edge of type 'shadow' is established from a shading object node $v_i$ (building or tree) with height $h_{obj}$ to a ground node $v_j$ if their Euclidean distance $d(v_i, v_j)$ is less than or equal to the shadow length $L_{shadow,t}$, and the angular deviation falls within a predefined shadow angle width $\Delta\varphi_{shadow}$. The shadow properties are calculated as follows:

$$L_{shadow,t} = h_{obj} / \tan(\theta_{elev,t}) \tag{3}$$

$$\varphi_{shadow,t} = (\varphi_{azimuth,t} + 180°) \bmod 360° \tag{4}$$

where $\theta_{elev,t}$ is the solar elevation angle at timestep t, $\varphi_{azimuth,t}$ is the solar azimuth angle, and $\varphi_{shadow,t}$ is the principal direction of the shadow projection.

VEGETATION EVAPOTRANSPIRATION. This edge type models localized bio-physical interactions, serving as a solar-dependent spatial filter. Specifically, a directed edge is established from a tree node $v_i$ to any other node $v_j$ if their Euclidean distance $d(v_i, v_j)$ does not exceed a dynamic

radius of influence, $R_{activity,t}$. This radius is calculated based on the global horizontal radiation $I_t$(Wh/m$^2$) for the current hour, where $R_{base}$ is a presettable base radius:

$$R_{activity,t} = R_{base} \cdot \text{clip}(I_t/1000, 0.5, 1.2) \tag{5}$$

CONVECTIVE DIFFUSION. To encode fluid dynamic anisotropy, this edge redefines topological proximity using a wind-modulated 'effective distance' metric (Eq. 7), approximating the advection process on a discrete graph. The condition for creating this edge is that their 'effective distance'$d_{eff}(v_i, v_j)$, must be less than or equal to a base local radius, $R_{local}$. This effective distance is adjusted by a modulation factor, $\alpha_{wind,t}$, which accounts for the wind speed $v_{wind,t}$ and wind direction alignment $\Delta\theta_{wind}$:

$$\alpha_{wind,t} = 1.0 + \lambda_{wind} \cdot \cos(\Delta\theta_{wind}) \cdot (v_{wind,t}/v_{max}) \tag{6}$$

$$d_{eff}(v_i, v_j) = d(v_i, v_j)/\alpha_{wind,t} \leq R_{local} \tag{7}$$

where $\lambda_{wind}$ is the wind effect intensity coefficient, determining the extent to which wind speed and direction stretch or compress the 'effective connection distance'.$v_{max}$ represents the maximum wind speed observed in the study scenario, ensuring the numerical stability of the model. These threshold parameters serve as preset physical upper bounds derived from established urban physics literature, ensuring the graph topology remains within a valid physical domain while allowing the network to learn specific interaction strengths. Detailed parameter configurations for constructing all edge types are provided in Appendix C.

**Static Semantic Topology**. To complement the dynamic physical edges and provide structural redundancy against heuristic imperfections, we introduce two static edge types that capture non-local dependencies and local continuity:

SEMANTIC SIMILARITY EDGES. To capture non-local functional interactions, we construct directed edges from each node to its $k$ nearest neighbors in the normalized static feature space. This mechanism allows the GNN to aggregate information from spatially distant but physically similar entities (e.g., similar materials), serving as a backup information pathway.

INTERNAL CONTIGUITY EDGES. To model intra-object energy transfer (thermal inertia) within large continuous bodies, 'internal nodes' establish connections with their eight immediate neighbors (Moore neighborhood). This ensures local physical consistency within building clusters or vegetation patches.

## 4.2 URBANGRAPH ARCHITECTURE

To align neural computation with the encoded structural priors, we designed a dynamic and heterogeneous architecture for UrbanGraph. As illustrated in Figure 3, the overall architecture comprises four core components: Feature Encoders, a Spatial Graph Encoder, a Spatio-Temporal Evolution Module, and a Prediction Head.

**Feature Encoders and Spatial Graph Encoder**. At timestep $t$, MLP-encoded global environmental ($u_t^{env}$) and temporal ($u_t^{time}$) features are broadcast and concatenated with spatial node representations $h_{v,t}^{RGCN}$ extracted by a three-layer RGCN. This fusion enables the subsequent LSTM to contextualize global forcing within local topologies. Crucially, the RGCN functions as a physical operator approximator: by assigning dedicated weight matrices $W_r$ (Eq. 2) to specific relations (e.g., shading vs. convection), it disentangles complex dynamics into distinct physical sub-processes, structurally mitigating the over-smoothing issues typical of homogeneous GNNs.

**Spatio-Temporal Evolution Module**. This module is responsible for fusing the spatial and global dynamic features and uses a Long Short-Term Memory (LSTM) network to model their temporal evolution. We specifically select LSTM over Attention mechanisms to align with the Markovian nature of physical transport processes (e.g., heat diffusion), where the future state evolves continuously from the immediate past. Furthermore, unlike coupled spatio-temporal operators (e.g., LRGCN), our architecture decouples spatial interaction from temporal evolution. By resolving the dynamic topology explicitly within the RGCN module, we provide the LSTM with a physics-consistent state representation, thereby reducing optimization complexity.

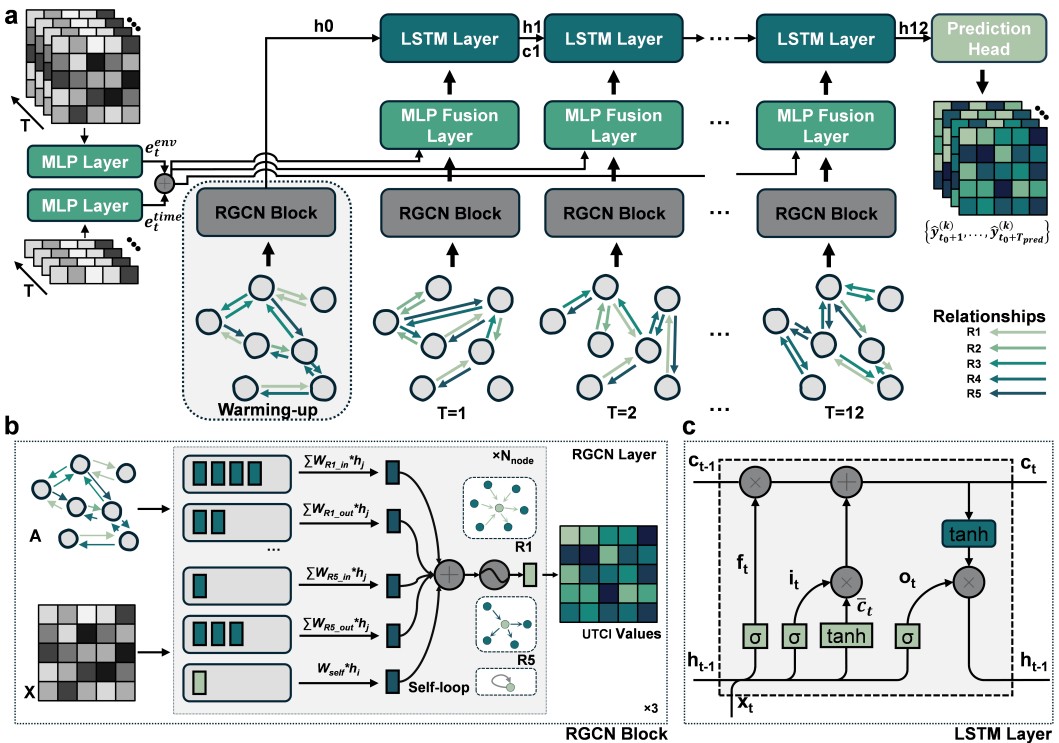

Figure 3: Architecture of UrbanGraph. (a) End-to-End Framework: Processes historical data, weather, and dynamic graphs. The top-left grids denote global and cyclically encoded time features. At each step, RGCN blocks extract spatial features, fused via MLP before LSTM temporal propagation. (b) RGCN Block: Aggregates multi-relational neighbor messages ($R1 - R5$) with node self-features. (c) LSTM Layer: Captures temporal dependencies.

At each prediction timestep t (from $t_1$ to $T_{pred}$), we concatenate the spatial representation of a node, $\boldsymbol{h}_{v,t}^{RGCN}$, with the global environmental embedding, $\boldsymbol{e}_t^{env}$, and the temporal embedding, $\boldsymbol{e}_t^{time}$. The resulting concatenated vector is passed through a fusion MLP to generate the input feature for the LSTM layer, $\boldsymbol{x}_{v,t}^{LSTM}$. This is expressed as:

$$\boldsymbol{x}_{v,t}^{LSTM} = \text{MLP}_{fusion}([\boldsymbol{h}_{v,t}^{RGCN} \oplus \boldsymbol{e}_t^{env} \oplus \boldsymbol{e}_t^{time}]) \tag{8}$$

The sequence of fused features is then fed into an LSTM layer to model the temporal dynamics. To provide the model with an effective initial state, an MLP projects the spatial features from the initial graph, $\boldsymbol{h}_{v,t_0}^{RGCN}$, to form the initial hidden state $\boldsymbol{h}_0$. The initial cell state $\boldsymbol{c}_0$ is initialized as a zero vector. This is expressed as:

$$\boldsymbol{h}_0 = \text{MLP}_{h_0}(\boldsymbol{h}_{v,t_0}^{RGCN}) \tag{9}$$

**Prediction Head**. Finally, a separate MLP decodes the last hidden state of the LSTM, $\boldsymbol{h}_{v,T_{pred}}^{LSTM}$, into a multi-step prediction vector, $\hat{\boldsymbol{y}}_v$. This generates the predictions for all $T_{pred}$ future timesteps at once. This is expressed as:

$$\hat{\boldsymbol{y}}_v = [\hat{y}_{v,1}, \ldots, \hat{y}_{v,Tpred}] = \text{MLP}_{head}(\boldsymbol{h}_{v,T_{pred}}^{LSTM}) \tag{10}$$

## 5 EXPERIMENTS

**Dataset**. Experiments utilize a high-fidelity ENVI-met dataset where the environment is discretized into a 3D grid (4m horizontal, 3m vertical) derived from high-precision vector and land cover data (Appendix B.1). While primarily evaluating UTCI, we predict all six variables to demonstrate the architecture's scalability. Temporal graphs follow principles in Section 4.1. The 396 urban blocks

are split into Training (70%), Validation (20%), and Testing (10%). Crucially, the test set comprises spatially distinct, unseen blocks to strictly evaluate generalization to new configurations.

**Baseline Models**. We benchmark UrbanGraph against four state-of-the-art categories: 1) Grid-based Methods (CGAN-LSTM (Isola et al., 2017), Pix2Pix+PINN) to assess graph necessity, where the latter utilizes soft physics-loss constraints; 2) Static Spatio-Temporal GNNs (GCN/GINE-LSTM, STGCN (Yu et al., 2018), ASTGCN (Guo et al., 2019)) to validate dynamic topology against fixed structures; 3) Generative Graph Models (GGAN-LSTM, GAE-LSTM) to compare with latent structure learning; and 4) Dynamic Graph Models (LRGCN (Li et al., 2019a)) to demonstrate the efficiency of our explicit causal encoding over implicit recurrent learning.

**Evaluation Metrics**. We evaluate predictive performance using Mean Absolute Error (MAE), Root Mean Squared Error (RMSE), and Coefficient of Determination ($R^2$). Computational efficiency is assessed via floating-point operations (FLOPs), training time, and inference speed. Models are trained using Adam with Mean Squared Error (MSE) loss—augmented by a KL divergence term for GAE and binary cross-entropy for CGAN. Training employs ReduceLROnPlateau scheduling and early stopping for robustness.

**Model Settings**. In the main comparative analysis, our proposed spatio-temporal heterogeneous model is configured with a learning rate of 0.001, a batch size of 8, a hidden dimension of 128 for all layers, a 3-layer RGCN encoder, and a 1-layer LSTM. It uses a multi-head prediction architecture, and all models are run for 3 independent trials. For the subsequent ablation studies and sensitivity analyses, we use a model with hyperparameters optimized by Optuna (Akiba et al., 2019), featuring a hidden dimension of 384 and a single prediction head. More detailed hyperparameter settings are available in Appendix D. All experiments were conducted on a single NVIDIA L4 GPU.

## 6 RESULT

We evaluate UrbanGraph by strictly benchmarking it against baselines in Section 6.1 and conducting core ablation studies in Section 6.2. Extended analyses, including additional ablations, hyperparameter sensitivity, and computational efficiency, are detailed in Appendix E.

### 6.1 MODEL PERFORMANCE

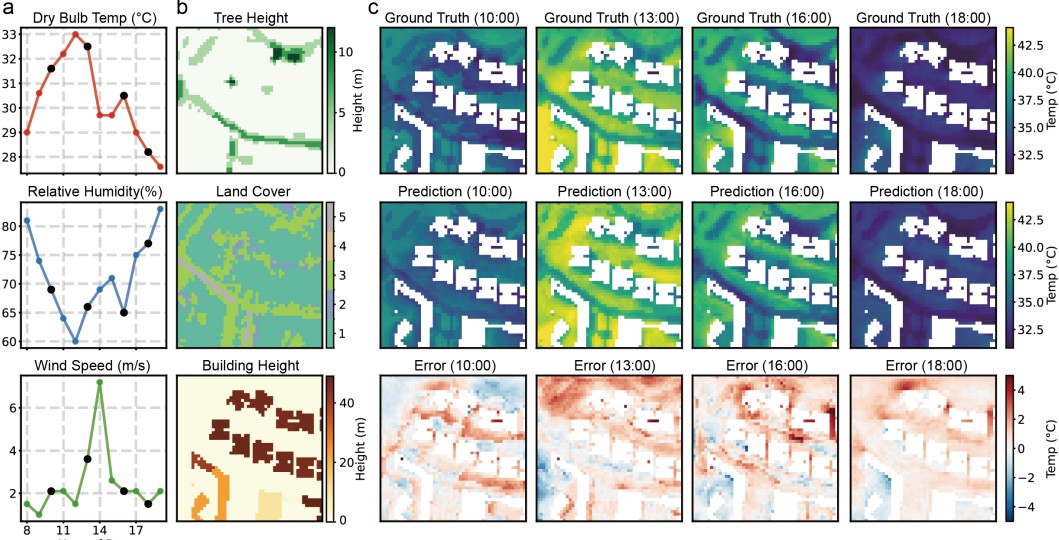

Figure 4: Input data configuration and prediction visualization. (a) Dynamic Global Features: Time-varying meteorological forcing (e.g., Wind Speed, Temperature, Relative Humidity). (b) Static GIS Features: Node-level spatial attributes including Building Height, Tree Height, and Land Cover ID. (c) Prediction Results: Visual comparison between Ground Truth, Model Prediction, and Absolute Error maps.

By integrating dynamic meteorological forcing with static GIS features, UrbanGraph achieves high-fidelity microclimate prediction across complex urban environments. As shown in Figure 4, visual comparisons confirm that the model accurately captures fine-grained thermal gradients, exhibiting strong spatial consistency with ground truth and minimal prediction error. To further substantiate these visual findings, we provide extended qualitative visualizations in Appendix F, which offer additional intuitive evidence of the model's ability to capture complex spatial distributions.

Table 1 shows UrbanGraph achieves state-of-the-art performance (highest $R^2 = 0.8542$). Compared to the strongest dynamic graph baseline LRGCN ($R^2 = 0.8422$), UrbanGraph improves accuracy while reducing FLOPs by 73.8% ($9.13 \times 10^9$ vs. $3.49 \times 10^{10}$) and training time by 21% (24.5s vs. 31.1s), validating the efficiency of explicit causal pruning over implicit recurrence. Furthermore, surpassing Pix2Pix+PINN (0.8320) confirms that hard structural constraints offer better physical consistency than soft loss constraints, a finding corroborated by ablation studies in Appendix E.

Table 1: Performance and efficiency comparison of different model architectures on the test set.

| Category | Model | Flops | Test | | | Time Cost | |
|---|---|---|---|---|---|---|---|
| | | | Avg $R^2$ | Avg RMSE | Avg MAE | Training (epoch/s) | Inference/s |
| Grid-based | CGAN-LSTM | $1.10 \times 10^{10}$ | $0.7712 \pm .0369$ | $1.3450 \pm .1175$ | $0.9539 \pm .0611$ | $15.3252 \pm 1.0999$ | $1.5558 \pm .1951$ |
| | Pix2Pix+PINN | $1.10 \times 10^{10}$ | $0.8320 \pm .0046$ | $1.1485 \pm .0306$ | $0.8365 \pm .0185$ | $17.5020 \pm 0.0576$ | $1.3031 \pm .0253$ |
| Static STGNNs | GCN-LSTM | $8.28 \times 10^9$ | $0.8347 \pm .0039$ | $1.1327 \pm .0433$ | $0.8544 \pm .0308$ | $28.5321 \pm 2.8358$ | $2.8619 \pm .4516$ |
| | GINE-LSTM | $8.80 \times 10^9$ | $0.8087 \pm .0226$ | $1.2045 \pm .0294$ | $0.9030 \pm .0190$ | $32.3169 \pm 1.4643$ | $3.1731 \pm .2325$ |
| | STGCN | $\mathbf{4.13 \times 10^9}$ | $0.7880 \pm .0065$ | $1.2958 \pm .0287$ | $0.9879 \pm .0243$ | $\mathbf{5.3547 \pm 0.0222}$ | $\mathbf{0.2928 \pm .0041}$ |
| | ASTGCN | $2.06 \times 10^{10}$ | $0.8317 \pm .0048$ | $1.1491 \pm .0234$ | $0.8579 \pm .0120$ | $25.4938 \pm 0.0438$ | $1.2903 \pm .0284$ |
| Generative Graph | GAE-LSTM | $1.05 \times 10^{10}$ | $0.8494 \pm .0036$ | $1.0687 \pm .0269$ | $0.7968 \pm .0128$ | $36.7376 \pm 3.2079$ | $3.6022 \pm .4504$ |
| | GGAN-LSTM | $9.44 \times 10^9$ | $0.8415 \pm .0034$ | $1.0981 \pm .0406$ | $0.8214 \pm .0319$ | $42.4678 \pm 3.1537$ | $2.6488 \pm .4073$ |
| Dynamic Graph | RGCN-GRU | $7.12 \times 10^9$ | $0.8483 \pm .0035$ | $1.0682 \pm .0380$ | $0.8020 \pm .0293$ | $20.8096 \pm 1.3612$ | $2.1640 \pm .2133$ |
| | RGCN-Transformer | $5.09 \times 10^{10}$ | $0.8465 \pm .0065$ | $1.0791 \pm .0253$ | $0.8066 \pm .0118$ | $37.6463 \pm .8325$ | $3.3345 \pm .1482$ |
| | LRGCN | $3.49 \times 10^{10}$ | $0.8422 \pm .0061$ | $1.0889 \pm .0321$ | $0.8100 \pm .0160$ | $31.0808 \pm 0.1877$ | $3.0550 \pm .0348$ |
| | **URBANGRAPH** | $9.13 \times 10^9$ | $\mathbf{0.8542 \pm .0044}$ | $\mathbf{1.0535 \pm .0338}$ | $\mathbf{0.7866 \pm .0250}$ | $24.4823 \pm 0.9323$ | $2.6914 \pm .1404$ |

The convergence curve (Figure 5a) confirms the stability of the model's training process. Moreover, the hour-by-hour error analysis (Figure 5b) shows that our method consistently maintains the lowest RMSE throughout the entire 12-hour prediction horizon. It demonstrates strong robustness against error accumulation, particularly during afternoon hours (e.g., 14:00 and 17:00) when climate fluctuations are more pronounced.

Real-world generalization requires dynamic edge rules that can withstand observational noise. We verified this robustness through extensive multi-scale validation (Appendix B.2.1). The model achieved high accuracy in both micro-scale calibration on the NUS campus ($r > 0.73$) and city-scale deployment across Singapore ($r = 0.842$).This successful generalization confirms that our parameterization strategy remains effective across heterogeneous urban morphologies.

To rigorously evaluate the model's generalization across distinct physical domains, we constructed the UWF3D dataset, which records high-resolution vector fields governed by Navier-Stokes equations. UrbanGraph demonstrates exceptional adaptability to this new physics, achieving high accuracy ($R^2 > 0.88$ for the $u$-component) and significantly outperforming the Grid-GCN baseline. This successful transition from scalar thermal diffusion to complex vector flow dynamics confirms the robustness of our explicit topological encoding framework in capturing diverse physical mechanisms. Detailed setup and results are provided in Appendix G.

## 6.2 ABLATION ANALYSIS

**Heterogeneous Graph Mechanism.** To validate the importance of modeling diverse physical interactions with distinct relation types, we compare our full model (Base), which uses a heterogeneous graph (RGCN), against a variant that simplifies the graph to be homogeneous (GCN). The results, shown in Table 2a, reveal a significant performance degradation when heterogeneity is removed, with the R² score dropping from 0.8629 to 0.8347. The performance drop highlights causal entangle-

Table 2: Ablation studies for key mechanisms.

(a) Heterogeneous.

| Model | R² | MSE |
|---|---|---|
| **Base** | **0.8629** | **1.0976** |
| Homo | 0.8336 | 1.4275 |

(b) Dynamic.

| Model | R² | MSE |
|---|---|---|
| **Base** | **0.8629** | **1.0976** |
| Static | 0.8057 | 1.6678 |

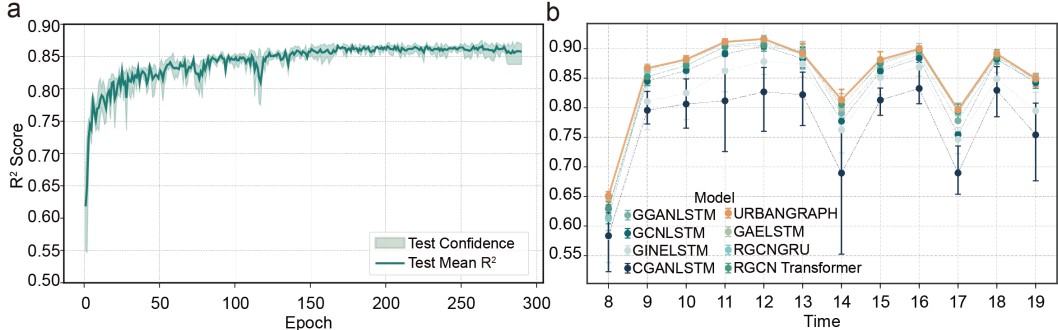

Figure 5: Model performance analysis. (a) Test set $R^2$ convergence curves for the UrbanGraph model. Shaded areas represent the confidence interval. (b) Hour-by-hour $R^2$ comparison between UrbanGraph and baselines on the test set, with error bars indicating standard deviation.

ment in homogeneous graphs where a single parameter set fits distinct physical laws. Heterogeneity resolves this by enabling physical operator decoupling.

**Dynamic Graph Mechanism**. To validate the effectiveness of the dynamic graph mechanism, we compare our model with a variant that uses a static graph (i.e., the same graph structure is shared across all timesteps). As shown in Table 2b, disabling the dynamic mechanism leads to a significant performance drop in the model (Static), with the R² score decreasing from 0.8629 to 0.8057. The gain validates time-varying causal pruning: unlike static graphs, the dynamic mechanism actively removes physically irrelevant connections (e.g., shifting shadows), thereby reducing optimization difficulty.

Further ablation studies analyzing other key components—such as the contribution of individual edge types, various prediction head architectures, feature fusion strategies, and the effects of explicit edge features—are detailed in Appendix E.

## 7 CONCLUSION

In this paper, we proposed UrbanGraph, a physics-informed dynamic graph framework for microclimate prediction. Achieving state-of-the-art accuracy ($R^2 = 0.8542$), UrbanGraph outperforms the strongest baseline (LRGCN) with a 73.8% reduction in FLOPs and 21% faster training. This result validates the superior efficiency of explicit causal pruning over implicit recurrence.

Furthermore, the UMC4/12 dataset, which we constructed and released, serves as the first high-resolution benchmark in this field and will help accelerate the development and fair comparison of new algorithms in the future. In summary, UrbanGraph demonstrates the potential of structural priors in bridging the gap between numerical simulation and deep learning. Beyond microclimate prediction, our work offers a generalizable explicit topological encoding paradigm for spatio-temporal dynamics governed by known physical equations.

**Limitation and Future Work**. Our work explicitly encodes predefined physical processes (i.e., prior knowledge) into the graph topology. While this has shown performance advantages, it may oversimplify the real physical processes, as it might overlook latent relationships present in the data that we have not yet modeled or are unknown. To address this trade-off between physical consistency and model flexibility, a critical future direction is to develop a Hybrid Topology framework. By augmenting the fixed physical graph with a sparse, learnable Residual Graph, future models could uncover unmodeled interaction patterns from data residuals. This approach effectively balances the robustness of engineering rules with the flexibility required to identify new physical mechanisms.

## REPRODUCIBILITY STATEMENT

To ensure reproducibility, our code and the UMC4/12 dataset are publicly available at `https://github.com/wlxin-nus/UrbanGraph.git`.

## ACKNOWLEDGMENTS

The authors acknowledge the use of a large language model for assistance with language editing and improving the clarity of this manuscript.

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

# A   KEY PHYSICAL EQUATIONS IN ENVI-MET

This appendix outlines the key physical equations within the ENVI-met model (Bruse & Fleer, 1998) used to generate the dataset for this study.

## A.1   MEAN AIR FLOW

The model describes three-dimensional turbulence by solving the non-hydrostatic, incompressible Navier-Stokes equations. The fundamental equations for the mean wind velocity components u,v,w are as follows:

$$\frac{\partial u}{\partial t} + u_i \frac{\partial u}{\partial x_i} = -\frac{\partial p'}{\partial x} + K_m \left(\frac{\partial^2 u}{\partial x_i^2}\right) + f(v - v_g) - S_u \tag{11}$$

$$\frac{\partial v}{\partial t} + u_i \frac{\partial v}{\partial x_i} = -\frac{\partial p'}{\partial y} + K_m \left(\frac{\partial^2 v}{\partial x_i^2}\right) - f(u - u_g) - S_v \tag{12}$$

$$\frac{\partial w}{\partial t} + u_i \frac{\partial w}{\partial x_i} = -\frac{\partial p'}{\partial z} + K_m \left(\frac{\partial^2 w}{\partial x_i^2}\right) + g\frac{\theta(z)}{\theta_{ref}(z)} - S_w \tag{13}$$

where $p'$ is the local pressure perturbation, $\theta$ is the potential temperature, $K_m$ is the turbulent diffusivity for momentum, $f$ is the Coriolis parameter, and $S_{u(i)}$ are the momentum source/sink terms induced by elements such as vegetation.

## A.2   TEMPERATURE AND HUMIDITY

The distribution of potential temperature $\theta$ and specific humidity $q$ in the atmosphere is described by the advection-diffusion equations, which include internal source/sink terms:

$$\frac{\partial \theta}{\partial t} + u_i \frac{\partial \theta}{\partial x_i} = K_h \left(\frac{\partial^2 \theta}{\partial x_i^2}\right) + Q_h \tag{14}$$

$$\frac{\partial q}{\partial t} + u_i \frac{\partial q}{\partial x_i} = K_q \left(\frac{\partial^2 q}{\partial x_i^2}\right) + Q_q \tag{15}$$

where $K_h$ and $K_q$ are the turbulent exchange coefficients for heat and moisture, respectively. $Q_h$ and $Q_q$ are the source/sink terms that couple the heat and moisture exchange processes at the surface and with vegetation.

## A.3   RADIATIVE FLUXES

The model solves the energy balance for surfaces and walls by calculating the net shortwave radiation, $R_{sw,net}$, and the net longwave radiation, $R_{lw,net}$. The shortwave radiation flux at any point, $R_{sw}(z)$, consists of direct and diffuse radiation, and accounts for the shading effects of buildings and vegetation:

$$R_{sw}(z) = \sigma_{sw,dir}(z)R_{sw,dir}^0 + \sigma_{sw,dif}(z)\sigma_{svf}(z)R_{sw,dif}^0 + (1 - \sigma_{svf}(z))R_{sw,dif}^0 \bar{\alpha} \tag{16}$$

where the $R^0$ terms represent the incoming radiation at the top of the model, and the $\sigma$ coefficients are reduction factors ranging from 0 to 1 that quantify the effects of direct radiation $\sigma_{sw,dir}$, diffuse radiation $\sigma_{sw,dif}$, and the sky view factor $\sigma_{svf}$.

For the complete set of model equations, parameterization schemes, and numerical solution methods, please refer to the original publication.

# B   HIGH-RESOLUTION SPATIO-TEMPORAL DATASET FOR MICROCLIMATE AND THERMAL COMFORT

## B.1   DATASET GENERATION

We constructed the UMC4/12 dataset based on public geospatial data and the ENVI-met model. We selected a typical extreme heat day as the basis for our simulations, using the standard meteorological year data (EPW) from Singapore Changi Airport. To ensure morphological diversity in the

dataset, we employed a stratified sampling strategy to select 11 representative 1 km² sites across Singapore. The stratification was based on key urban morphology metrics, and the sample pool covers a wide range of urban typologies, from ultra-high-density commercial districts to mature residential areas with large parks (see Appendix Table A1). The metrics include Average Building Height (Avg.BH), Green Space Ratio (GSR), and Building Coverage Ratio (BCR).

Table A1: Distribution of morphological and material properties for the 11 selected 1km² sites in Singapore.

| Data Index | Avg.BH(m) | GSR | BCR | Pavement% | Smashed Brick% | Loamy Soil% | Deep Water% |
|---|---|---|---|---|---|---|---|
| 1 | 13.36 | 0.021 | 0.078 | 0.520 | 0.095 | 0.367 | 0.019 |
| 2 | 19.76 | 0.055 | 0.155 | 0.734 | 0.062 | 0.181 | 0.023 |
| 3 | 12.86 | 0.255 | 0.219 | 0.487 | 0.000 | 0.471 | 0.043 |
| 4 | 23.97 | 0.184 | 0.217 | 0.630 | 0.043 | 0.316 | 0.010 |
| 5 | 10.11 | 0.116 | 0.235 | 0.643 | 0.026 | 0.302 | 0.029 |
| 6 | 12.01 | 0.429 | 0.126 | 0.260 | 0.020 | 0.718 | 0.002 |
| 7 | 28.14 | 0.165 | 0.242 | 0.671 | 0.023 | 0.286 | 0.020 |
| 8 | 13.86 | 0.209 | 0.338 | 0.733 | 0.036 | 0.220 | 0.011 |
| 9 | 33.73 | 0.198 | 0.108 | 0.297 | 0.050 | 0.554 | 0.098 |
| 10 | 19.81 | 0.105 | 0.109 | 0.291 | 0.023 | 0.222 | 0.464 |
| 11 | 16.06 | 0.444 | 0.128 | 0.211 | 0.062 | 0.654 | 0.072 |

We built the 3D model input files for the ENVI-met simulations by integrating multiple public geospatial data sources. Specifically, we resampled and performed 3D voxelization on raw data with varying precisions: building footprints were extracted from vector-based OpenStreetMap data (Haklay & Weber, 2008), while the land cover classification (Gaw et al., 2019) and canopy height maps (Tolan et al., 2024) utilized a 1m high resolution. This process generated ENVI-met input files (.INX) with a uniform horizontal resolution of 4 meters and a vertical resolution of 3 meters. To ensure high fidelity, we assigned realistic material properties to different surfaces and building boundaries, and specified corresponding tree species for vegetation of varying heights. The detailed material assignments and parameters are provided in Appendix Table A2 and A3. The simulation period covered the hours from 08:00 to 19:00, when urban heat effects are most significant.

Table A2: Class definitions mapping land cover types to surface materials for ENVI-met simulation.

| Type | Material | Class |
|---|---|---|
| Buildings | Pavement | 1 |
| Impervious surfaces | Pavement | 1 |
| Non-vegetated pervious surfaces | Terre battue | 2 |
| Vegetation with limited human management (w/ Tree Canopy) | Loamy Soil | 3 |
| Vegetation with limited human management (w/o Tree Canopy) | Loamy Soil | 3 |
| Vegetation with structure dominated by human management (w/ Canopy) | Loamy Soil | 3 |
| Vegetation with structure dominated by human management (w/o Canopy) | Loamy Soil | 3 |
| Freshwater swamp forest | Unsealed Soil | 4 |
| Freshwater marsh | Unsealed Soil | 4 |
| Mangrove | Deep Water | 5 |
| Water courses | Deep Water | 5 |
| Water bodies | Deep Water | 5 |
| Marine | Deep Water | 5 |

Figure A1 provides a visualization of the primary input data layers—tree height, land cover type, and building height—for two representative sites, illustrating the morphological diversity within the UMC4/12 dataset.

Following the ENVI-met simulation, we generated high-resolution spatio-temporal data for the six target variables. Figure A2 illustrates the simulation output for one of the urban blocks, displaying the evolution of all six variables over the course of the day.

To expand the dataset while efficiently managing computational resources, we systematically partitioned the original 1 km² simulation results into 250m × 250m blocks with a 50-meter overlapping

Table A3: Material properties used in the ENVI-met model configuration.

| Material | $z_0$ Roughness Length | Albedo | Emissivity |
|---|---|---|---|
| Pavement | 0.010 | 0.3 | 0.9 |
| Terre battue | 0.010 | 0.4 | 0.9 |
| Loamy Soil | 0.015 | 0.0 | 0.9 |
| Unsealed Soil | 0.015 | 0.2 | 0.9 |
| Deep Water | 0.010 | 0.0 | 0.9 |

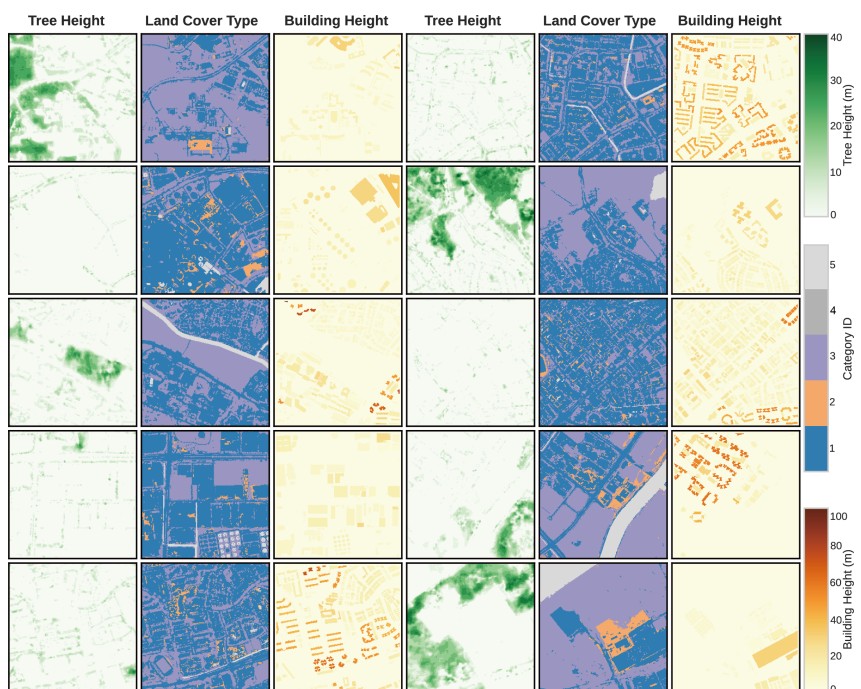

Figure A1: Visualization of the input data for ten sample sites from the UMC4/12 dataset. For every site, three data layers are visualized: (left in the triplet) Tree Height, (middle) Land Cover Type, and (right) Building Height.

area. This resulted in a final dataset containing 396 unique urban blocks. Each block is discretized into 2,500 nodes. For each block, we provide a time series covering a 12-hour interval for 6 key microclimate and thermal comfort variables. Overall, the UMC4/12 dataset offers approximately 11.9 million high-quality spatio-temporal data points for each target variable, enabling the systematic evaluation of spatio-temporal prediction models in complex urban environments.

## B.2 REAL-WORLD VALIDATION STRATEGY

To ensure that our physics-informed framework generalizes to real-world scenarios, we implemented a two-tier validation process covering both micro-scale fidelity and city-scale consistency.

### B.2.1 MICRO-SCALE CALIBRATION WITH FIELD MEASUREMENTS

We conducted a rigorous validation of the ENVI-met model based on a sensor network deployed on the campus of the National University of Singapore (NUS).

As shown in Figure A3a, a sensor on an open rooftop (Reference Point) was selected to provide the driving meteorological inputs. We compared simulation results with measured data from five distributed sensor stations covering diverse morphologies.

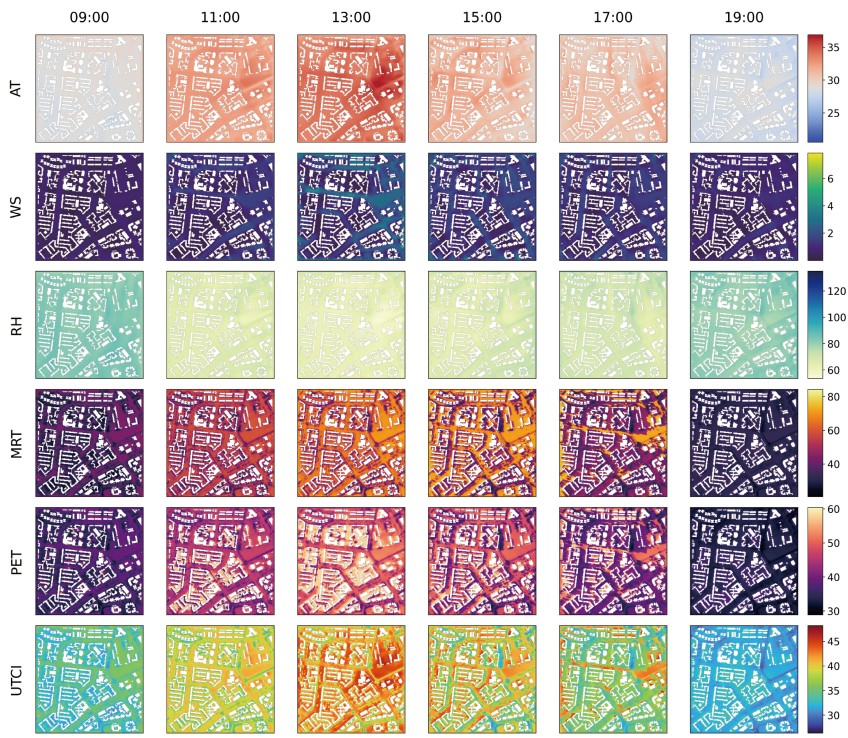

Figure A2: Visualization of the spatio-temporal simulation output for a single urban block. Each row corresponds to a different target variable: AT, WS, RH, MRT, PET, and UTCI. Each column represents a specific hour, showing the dynamic evolution of the microclimate from morning (09:00) to evening (19:00).

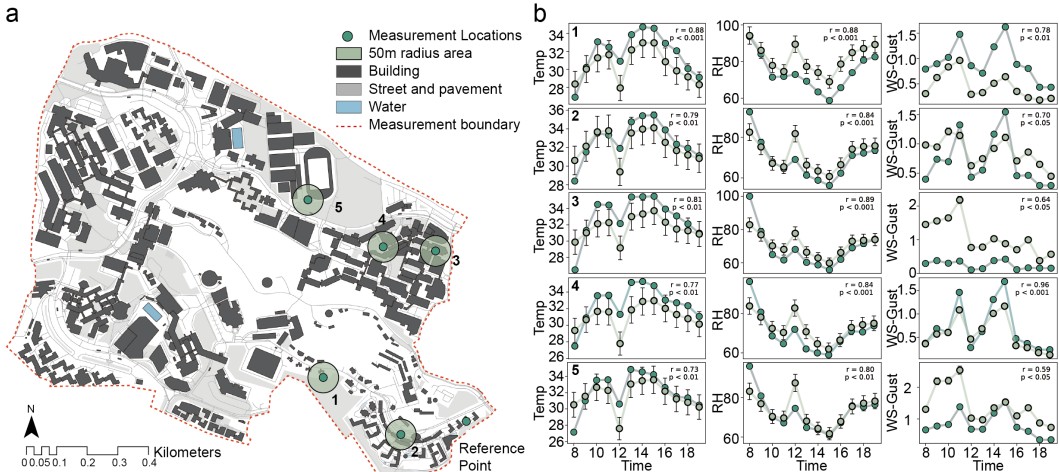

Figure A3: ENVI-met model validation. (a) Map of the validation area and sensor locations on the NUS campus. (b) Comparison of simulated vs. measured diurnal cycles for Temperature, Relative Humidity, and Wind Speed, showing strong agreement ($r > 0.73$).

The plots in Figure A3b show the hourly variations of key variables: air temperature, relative humidity, and gust wind speed. The results indicate high agreement with measured values (Pearson's correlation coefficient $r > 0.73$, $p < 0.01$), accurately capturing diurnal trends. This confirms that our simulation setup reliably reproduces key microclimate dynamics in complex urban environments, providing a solid physical basis for subsequent data-driven modeling.

### B.2.2  CITY-SCALE CROSS-VALIDATION

To further assess the model's dynamic response capability in the temporal dimension and its generalization at the city scale, we introduced the HiGTS dataset (Jian et al., 2024), which provides global hourly UTCI at a $0.1° \times 0.1°$ resolution.

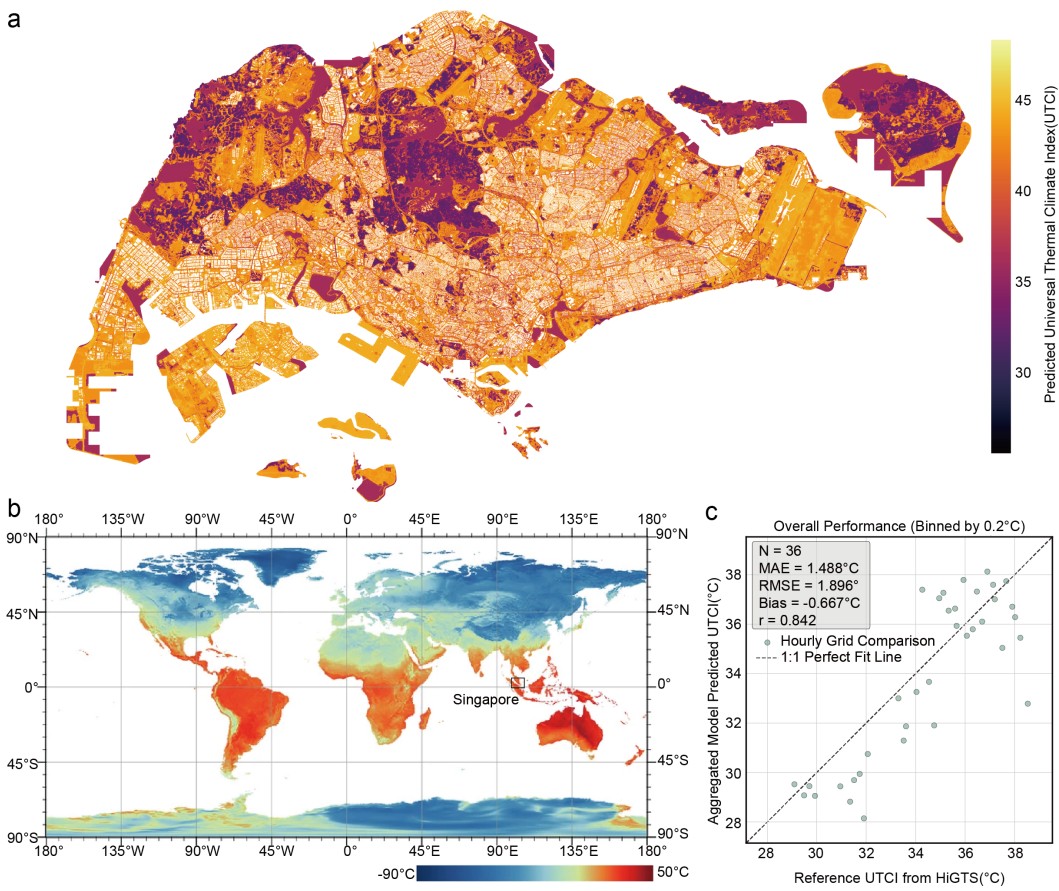

Figure A4: City-scale generalization validation against the HiGTS dataset. The scatter plot compares UrbanGraph predictions with HiGTS reference hourly UTCI for Singapore. The results demonstrate a strong correlation ($r = 0.842$) and low bias, confirming the model's robustness in generalizing to real-world urban environments.

We deployed the trained UrbanGraph model to predict the hourly UTCI for the entire Singapore island on May 3rd, 2023, and compared the aggregated results with the HiGTS reference values.

As shown in the scatter plot in Figure A4, the two datasets show a strong correlation ($r = 0.842$), with a slight systematic underestimation (Bias = $-0.667°C$). The hourly analysis reveals that prediction bias primarily originates from trend discrepancies between the simulation inputs and the reference dataset at specific moments (e.g., abrupt weather changes) rather than flaws in the model itself. This validates that UrbanGraph can effectively generalize to real-world urban data beyond the training distribution.

## C    Parameter Settings for Physics-Informed Graph Representation

This appendix details the key parameters used in the construction of the dynamic heterogeneous graph, as introduced in Section 4.1, and provides the rationale for their settings.

### C.1    Parameter Rationale

**Number of Nearest Neighbors (k).**    The value is chosen to balance informational richness with computational overhead. Allowing each node to connect to its eight most similar neighbors (consistent with the size of a Moore neighborhood) effectively captures non-local semantic information while avoiding the noise that could be introduced by connecting too many distant nodes. As demonstrated in the sensitivity analysis in Section E.2, this value represents the optimal trade-off between model performance and efficiency.

**Maximum Shadow Extent ($R_{max}^{shadow}$).**    These upper limits are set to prevent unrealistically long shadows, which can occur at low solar elevation angles, from creating computational redundancy in the graph representation. The maximum shadow extent for buildings (15 grids, or 60m) is larger than that for trees (5 grids, or 20m), which is consistent with their typical differences in height and obstruction capacity in an urban environment.

**Shadow Angle Width ($\Delta\phi_{shadow}$).**    This parameter expands the theoretical line-like shadow into an area of influence. This accounts for the apparent motion of the sun over an hour and the penumbra effect caused by diffuse light, making the shadow model more physically realistic.

**Base Radius of Influence for Vegetation ($R_{base}$).**    The base radius of influence for vegetation is set to 5 grids (20m), based on the typical effective range of local cooling effects from single or small patches of green space reported in existing microclimate research (Kim et al., 2024).

**Wind Effect Coefficient ($\lambda_{wind}$).**    As a modulation coefficient, a value of 0.3 is a relatively conservative choice. It allows the wind field to significantly guide the anisotropy of connections without completely dominating the graph structure, thus preserving the influence of other physical processes.

**Maximum Reference Wind Speed ($v_{max}$).**    This value is used to normalize the actual wind speed. A value of 8.0 m/s was chosen as the reference upper limit as it represents the maximum wind speed historically observed in Singapore.

The boundary parameters for these edges are structural constraints, with their values set by referencing classical urban climatology studies (Ziter et al., 2019; Dare, 2005; Ng, 2009; Lin et al., 2023). This setting is necessary for explicit physical embedding, as it translates domain knowledge (e.g., the effective decay distance of vegetation cooling) into reliable physical upper bounds for most typical urban environments. Within these bounds, the actual interaction strength is adaptively learned by the network. The robustness of this setup is empirically validated by its successful generalization to the UWF3D dataset (Appendix G), where these physical bounds yielded high accuracy on a completely different physical task.

## D    Model Implementation Details and Hyperparameters

To ensure fairness, transparency, and reproducibility in our experimental comparisons, this appendix details the implementation specifics and key hyperparameter configurations for our proposed Urban-Graph model and all baseline models.

### D.1    Baseline Models and Hyperparameter Settings

The following table summarizes the key hyperparameters for UrbanGraph and all baseline models used in the different experimental phases. In our comparative experiments, we strive to ensure a fair comparison by maintaining a similar model scale (i.e., hidden dimension size), such that performance differences primarily originate from the model architectures themselves.

Table A4: Parameters for Dynamic Heterogeneous Adjacency Construction.

| Parameter | Value | Description |
|---|---|---|
| *Semantic Similarity Links* | | |
| $k$ | 8 | The number of neighbors for semantic similarity links. |
| $\epsilon$ | 1e-6 | A small constant to avoid division by zero during feature normalization. |
| *Shadow Links* | | |
| $R_{max}^{shadow}$ | 15 | Maximum extent of building shadows (in number of grids). |
| $R_{max}^{tree}$ | 5 | Maximum extent of tree shadows (in number of grids). |
| $\Delta\phi_{shadow}$ | 25.0° | The effective angular width for shadow calculations. |
| *Vegetation Activity Links* | | |
| $R_{base}$ | 5 | The base maximum radius of influence for vegetation activity (in number of grids). |
| *Local Wind Field Links* | | |
| $\lambda_{wind}$ | 0.3 | Coefficient that modulates the impact of wind direction on the connection range. |
| $v_{max}$ | 8.0 m/s | Used to normalize wind speed for calculating the wind modulation factor. |

Table A5: Key hyperparameters for the proposed model and all baseline models.

| Model | Hidden Dim | Spatial Encoder | Temporal Encoder | Key Hyperparameters |
|---|---|---|---|---|
| UrbanGraph (Ours) | 128/384* | RGCN(3) | LSTM(1) | `lr=0.001, batch_size=8, optimizer=Adam, weight_decay=1e-5` |
| GCN-LSTM | 128 | GCN(3) | LSTM(1) | same |
| GINE-LSTM | 128 | GINE(3) | LSTM(1) | same |
| RGCN-GRU | 128 | RGCN(3) | GRU(1) | same |
| RGCN-Transformer | 128 | RGCN(3) | Transformer | `d_model=128, nhead=4, num_encoder_layers=2` |
| CGAN-LSTM | 128 | U-Net | LSTM(1) | `lr_G=0.0002, lr_D=0.0002, beta1=0.5, lambda_L1=100` |
| GAE-LSTM | 128 | GAE(3) | LSTM(1) | `latent_dim=128, beta=0.1` |
| GGAN-LSTM | 128 | GGAN | LSTM(1) | `latent_dim=128, lr_G=0.0001, lr_D=0.0004, beta1=0.5` |

*Note: The hidden dimension of UrbanGraph is 128 in the main model comparison phase. For the ablation and sensitivity analysis phases, it is set to 384 based on the results of Optuna optimization.*

## D.2 IMPLEMENTATION DETAILS FOR CROSS-PARADIGM BASELINES

To compare our graph-based approach with traditional grid-based methods, we adapted the data input for certain baseline models.

**Data Rasterization.** For the CGAN-LSTM model, we convert the graph data at each timestep into a 50x50 grid image. Each node in the graph is mapped to a pixel in the image, where the pixel value represents a key physical feature of the node (e.g., air temperature). The spatial relationships between nodes are implicitly represented by the adjacency of pixels on the 2D plane.

**Model Implementation.** We employ a classic U-Net as the generator for the CGAN and a Patch-GAN as the discriminator. The model's task is to generate the prediction image for the next timestep based on a sequence of historical images. During training, we combine an L1 loss (with weight $\lambda_{L1}$) with an adversarial loss. The feature sequence extracted by the U-Net encoder is then fed into an LSTM module for temporal modeling.

# E   DETAILED ANALYSIS OF MAIN EXPERIMENTS

## E.1   DETAILED ABLATION STUDIES

We conduct a series of ablation studies to systematically evaluate the contributions of the key components within the UrbanGraph framework.

**Physics Injection Strategy**.   To rigorously isolate the impact of different physics-injection paradigms, we conducted a controlled experiment. We compared three variants sharing the identical backbone architecture (GCN+LSTM) and hyperparameter settings (e.g., hidden dimensions, learning rate). The only difference lies in the mechanism of physics injection: 1) Baseline: A standard data-driven model without any physical constraints. 2) Soft Constraint: The baseline augmented with PDE-based loss functions (PINN approach). 3) Hard Constraint (Ours): The proposed UrbanGraph that encodes physics directly into the graph topology.

Table A6: Quantitative comparison of different physics injection strategies.

| Strategy | Flops | Test | | | Time Cost | |
|---|---|---|---|---|---|---|
| | | Avg R² | Avg RMSE | Avg MAE | Training (epoch/s) | Inference/s |
| Baseline (Data-driven) | $8.28 \times 10^9$ | $0.7846 \pm .0039$ | $1.2918 \pm .0389$ | $0.9953 \pm .0248$ | $\mathbf{11.9636 \pm 0.0046}$ | $\mathbf{1.0002 \pm .0082}$ |
| Soft Constraint (PINN) | $8.28 \times 10^9$ | $0.7876 \pm .0023$ | $1.3028 \pm .0531$ | $1.0037 \pm .0359$ | $12.0993 \pm 0.0127$ | $1.0164 \pm .0123$ |
| **Hard Constraint (Ours)** | $9.13 \times 10^9$ | $\mathbf{0.8542 \pm .0044}$ | $\mathbf{1.0535 \pm .0338}$ | $\mathbf{0.7866 \pm .0250}$ | $24.4823 \pm 0.9323$ | $2.6914 \pm .1404$ |

As shown in Table A6, the 'Soft Constraint' yields negligible improvement ($\Delta R^2 \approx +0.0030$) over the baseline. In contrast, the 'Hard Constraint' achieves a substantial performance leap ($\Delta R^2 \approx +0.0696$). Although constructing the dynamic topology incurs a computational cost (increasing training time from  12s to 24.5s), the return on investment is significant: The hard structural constraint delivers over 23 times the accuracy gain of the soft loss constraint.

**Temporal Modeling**.   To validate the contribution of the Spatio-Temporal Evolution Module (LSTM), we compare the full Spatio-Temporal model against a variant where the LSTM module is removed. This variant performs independent predictions for each hour, thereby eliminating temporal dependencies. As shown in Figure A5 in the Appendix, the results demonstrate the effectiveness of temporal modeling. Our model's predictive accuracy (R²) surpasses that of the variant across all prediction hours. Furthermore, Our model exhibits lower variance across multiple independent trials, indicating enhanced model stability.

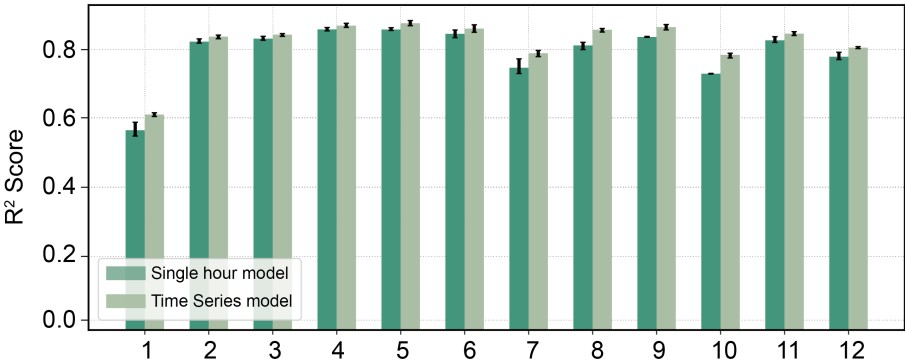

Figure A5: Hour-by-hour R² score comparison for the temporal modeling ablation study. The 'Time Series model' (our full UrbanGraph model) is compared against the 'Single hour model' (a variant without the LSTM module). The results demonstrate that explicitly modeling temporal dependencies leads to superior performance across the entire 12-hour prediction horizon. Error bars represent the standard deviation from multiple independent trials.

**Fusion Mechanism**. We compare three different strategies for fusing the spatial node representations with the global dynamic features: Concatenation Fusion, Multiplicative Fusion, and Attention

Fusion. As shown in Table A7, the simple concatenation strategy achieves the best performance across all evaluation metrics. This approach not only achieves the highest accuracy but also has the lowest computational load (FLOPs) and the fastest training and inference speeds.

Table A7: Comparison of different fusion strategies for combining spatial and global features. Best results are in **bold**.

| Strategy | Flops | Test | | Time Cost | |
|---|---|---|---|---|---|
| | | Avg R² | Avg RMSE | Training (epoch/s) | Inference/s |
| Attention | $9.42 \times 10^9$ | $0.8491 \pm .0052$ | $1.0675 \pm .0254$ | $27.5568 \pm 1.0310$ | $3.1741 \pm .1251$ |
| Multiplicative | $9.30 \times 10^9$ | $0.8515 \pm .0020$ | $1.0623 \pm .0317$ | $25.9573 \pm 1.4409$ | $2.8543 \pm .2170$ |
| Concatenation | $\mathbf{9.13 \times 10^9}$ | $\mathbf{0.8542 \pm .0044}$ | $\mathbf{1.0535 \pm .0338}$ | $\mathbf{24.4823 \pm 0.9323}$ | $\mathbf{2.6914 \pm .1404}$ |

**Prediction Head Architecture**. We evaluate two strategies for multi-step prediction: a Single-Head architecture, which uses a single shared prediction head to generate predictions for all 12 hours at once from the final hidden state of the LSTM; and a Multi-Head architecture, which employs a separate prediction head for each future timestep. The results in Table A8 show that the single-head strategy performs better in terms of both predictive accuracy and computational efficiency (FLOPs). For a relatively short prediction horizon, the single-head architecture can more effectively leverage the final hidden state, which encodes information from the entire sequence, for joint prediction, thereby avoiding cumulative errors.

Table A8: Comparison between Single-Head and Multi-Head prediction architectures on the test set.

| Strategy | Flops | Test | | Time Cost | |
|---|---|---|---|---|---|
| | | Avg R² | Avg RMSE | Training (epoch/s) | Inference/s |
| Multi-Head | $9.21 \times 10^9$ | $0.8542 \pm .0044$ | $1.0535 \pm .0338$ | $24.4823 \pm 0.9323$ | $2.6914 \pm .1404$ |
| Single-Head | $\mathbf{9.13 \times 10^9}$ | $\mathbf{0.8603 \pm .0008}$ | $\mathbf{1.0190 \pm .0421}$ | $\mathbf{21.5903 \pm 3.1143}$ | $\mathbf{2.4785 \pm .5019}$ |

**Warming-up Mechanism**. We introduce a warming-up mechanism that initializes the LSTM's hidden state using the spatial features from the initial graph. This aims to provide the temporal prediction task with a starting point that is rich in physical priors. As shown in Table A9, removing this mechanism and using random initialization instead (the NP1 model) leads to a noticeable decline in performance, with the R² score dropping from 0.8629 to 0.8510.

Table A9: Effectiveness of the Warming-up Mechanism.

| Model | R² | MSE |
|---|---|---|
| **Base** | **0.8629** | **1.0976** |
| NP1 | 0.8510 | 1.1526 |

**Node Feature Augmentation**. We compare the effects of using different node features as input. As shown in Table A10a, the model using aggregated neighbor features (Base) achieves the best performance. The model without any spatial information enhancement (M3) performs worse than the Base model. However, performance degrades when using only static topological features (such as degree centrality) or when combining them with aggregated neighbor features. This result suggests that introducing additional topological features in our task may add redundant information or noise, thereby impairing the model's predictive accuracy.

**Input Feature Ablation**. To verify the necessity of each input feature, we conduct a systematic ablation on the static node features $\boldsymbol{u}^{static}$, the temporal encoding features $\boldsymbol{u}_t^{time}$, and the global climate features $\boldsymbol{u}_t^{env}$. As shown in Table A10b, the baseline model (Base) that includes all three feature types performs the best. Removing any single feature type leads to a performance drop. Notably, the model using only static node features (F1) shows the most significant degradation, with its R² score dropping from 0.8629 to 0.7179.

**Edge Types**. To evaluate the specific contribution of each of the five proposed physics-informed and semantic edge types, we conduct an ablation study by systematically removing one edge type at a time. As shown in Table A11, the base model, which includes all five edge types, achieves the best performance. Removing any single edge type results in a decline in the model's predictive accuracy, demonstrating that both the physics-informed and semantic edges provide valuable

Table A10: Ablation studies for feature augmentation and input feature types.

(a) Data Augmentation.

| Model | Neighbor | Structure | R² | MSE |
|---|---|---|---|---|
| M1 | | ✓ | 0.8347 | 1.2798 |
| M2 | ✓ | ✓ | 0.8462 | 1.2181 |
| M3 | | | 0.8507 | 1.1696 |
| **Base** | ✓ | | **0.8629** | **1.0976** |

(b) Input Features.

| Model | $u^{static}$ | $u_t^{time}$ | $u_t^{env}$ | R² | MSE |
|---|---|---|---|---|---|
| F1 | ✓ | | | 0.7179 | 2.0867 |
| F2 | ✓ | ✓ | | 0.8529 | 1.1423 |
| F3 | ✓ | | ✓ | 0.8519 | 1.1495 |
| **Base** | ✓ | ✓ | ✓ | **0.8629** | **1.0976** |

inductive biases for the model. Notably, removing the *Local Wind* and *Shadow* edges leads to the most significant performance degradation, which underscores the importance of explicitly modeling time-varying physical processes. Furthermore, the performance drop caused by removing *Similarity* edges confirms the necessity of capturing non–local spatial interactions in urban microclimate prediction.

Table A11: Ablation study on different edge types. The checkmark ($\sqrt{}$) indicates that the corresponding edge type is included in the model.

| Model | Tree activity | Similarity | Shadow | Local Wind | Internal | R² | MSE |
|---|---|---|---|---|---|---|---|
| E1 | | ✓ | ✓ | ✓ | ✓ | 0.8504 | 1.1534 |
| E2 | ✓ | | ✓ | ✓ | ✓ | 0.8531 | 1.1568 |
| E3 | ✓ | ✓ | | ✓ | ✓ | 0.8238 | 1.4960 |
| E4 | ✓ | ✓ | ✓ | | ✓ | 0.8155 | 1.4341 |
| E5 | ✓ | ✓ | ✓ | ✓ | | 0.8425 | 1.2403 |
| **Base** | ✓ | ✓ | ✓ | ✓ | ✓ | **0.8629** | **1.0976** |

## E.2 SENSITIVITY AND COMPUTATION PERFORMANCE EVALUATION

**Sensitivity to the Number of Neighbors (k).** To investigate the model's sensitivity to the number of neighbors, k, used in constructing the Semantic Similarity Edges, we conducted tests with different values of k. As shown in Figure A6a, the model's performance (R²) improves as k increases, reaching a peak at k=8 before exhibiting minor fluctuations. Considering that a larger k increases graph density and computational cost, we select the 'elbow point' of the performance curve, k=8, as the optimal configuration. The model is not highly sensitive to the choice of k within a certain range, demonstrating good robustness.

**Sensitivity to Training Data Volume**. To evaluate the model's data efficiency and generalization capability, we performed a sensitivity test on the amount of training data. We reserved a fixed 10% test set and incrementally increased the training set size using fractions of the remaining data, starting from 2%. As illustrated in Figure A6b, the results reveal a significant positive correlation between model performance and data volume, with all accuracy metrics improving substantially as the amount of data increases. However, the model also exhibits a clear diminishing returns effect: the majority of the performance gain occurs before the training data volume reaches 40-60%, after which the performance curve begins to plateau. Performance tends to saturate when approximately 90% of the available training data is used.

**Computational Performance Evaluation**. To assess the model's computational overhead in urban scenarios of varying complexity, we analyzed the relationship between the graph's structural properties (i.e., the number of nodes and edges) and computational costs (inference time and peak GPU memory usage). The analysis (Figure A7b,d) indicates that both inference time and memory consumption show a positive correlation with the number of edges in the graph (with R² values of 0.5976 and 0.4627, respectively). An interesting finding is that computational cost is negatively correlated with the number of non-building nodes (Figure A7a,c). This suggests that the number of non-building nodes can serve as an inverse indicator of a scenario's structural complexity: scenes with more open spaces (e.g., parks) typically have sparser graph structures and are therefore more computationally efficient. Furthermore, the linear relationship between cost and graph complexity suggests the feasibility of applying the model to larger areas.

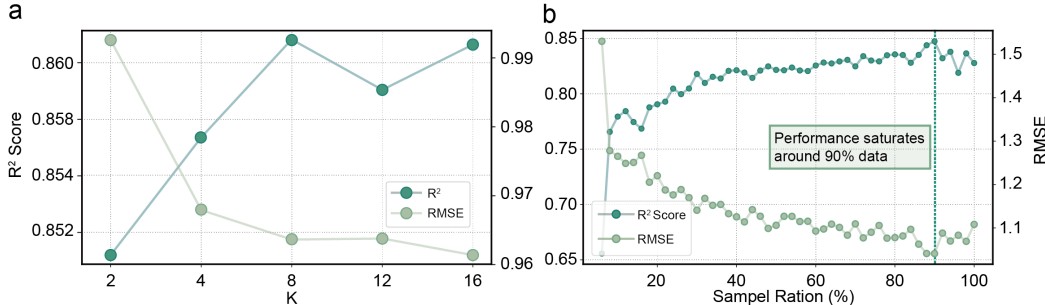

Figure A6: Sensitivity analysis of the model. (a) Model performance ($R^2$ and RMSE) on the test set versus the number of neighbors, k, for constructing similarity edges. The performance peaks at k=8. (b) Model performance as a function of the percentage of training data used. Performance gains show diminishing returns and begin to saturate at approximately 90% of the data.

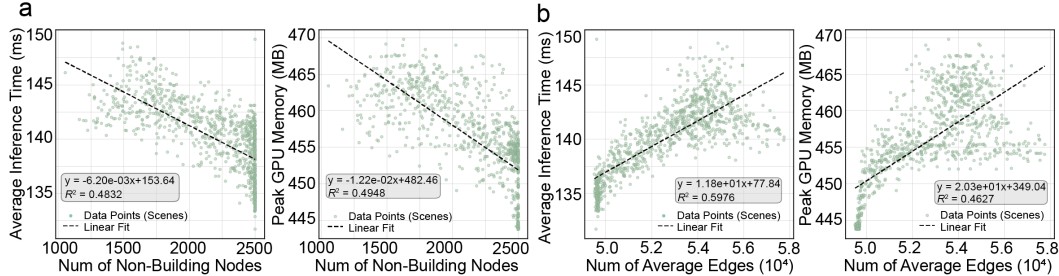

Figure A7: Computational performance analysis. (a) illustrate the negative correlation between inference time / peak GPU memory and the average number of non-building nodes per window. (b) show the positive linear correlation between computational costs and the average number of edges per window.

**Scalability Test on City-Scale Grids**. To verify the feasibility of deploying UrbanGraph on large-scale urban grids (as queried regarding 50k–100k nodes), we conducted a stress test on a large urban region. The region was partitioned into 36 spatial windows, comprising a total of approximately 90,000 nodes. The total edge reconstruction time for the entire 13-hour sequence was recorded at 39.56 seconds (using 16 CPU workers). This empirical evidence confirms that the computational cost for city-scale edge reconstruction is negligible compared to physics-based simulations, and the divide-and-conquer strategy effectively ensures scalability.

### E.3    EDGE FEATURES AND WEIGHTS

To explore the potential of encoding richer physical information into the graph structure, we designed and evaluated an explicit scheme for edge attributes and edge weights in the early stages of our research. As mentioned in the main text, our final model did not adopt this design, as experimental results showed that introducing this explicit information did not lead to a significant performance improvement for the UTCI prediction task. This section details our initial exploratory design.

EDGE ATTRIBUTE VECTOR.    In our initial design, each edge $e_{ij} \in \mathcal{E}_t$ in the graph carried a 5-dimensional attribute vector $\boldsymbol{a}_{ij} \in \mathbb{R}^5$ to encode rich spatio-temporal physical information. This vector was composed of the following components:

- **Euclidean Distance** $d(v_i, v_j)$: The straight-line distance between the nodes.

- **Relative Displacement** $(\Delta x, \Delta y)$: The difference in position in the grid coordinate system.

- **Wind Alignment** $(cos(\Delta\theta_{wind}))$: The cosine of the angle between the edge vector and the current hour's wind direction, used to quantify the convective influence of the wind.

- **Edge Type ID**: A categorical ID indicating which of the five relationship types the edge belongs to.

### E.4 EDGE WEIGHT CALCULATION

To quantify the interaction strength between different nodes, we designed a dynamic scheme for calculating edge weights, $w_{ij}$. All edge weights start from a base value $w_{base}$ (set to 1.0) and are dynamically modulated according to the following rule:

$$w_{ij} = w_{base}/(1 + \lambda \cdot d(v_i, v_j)/d_{grid}) \cdot \beta \cdot \gamma \qquad (17)$$

The specific settings for each modulation factor are as follows:

- **Distance Decay** ($\lambda$)**:** All edge weights are decayed based on their Euclidean distance. To better distinguish between non-local and local effects, we set a smaller distance decay factor, $\lambda_{sim}$ (set to 0.005), for semantic similarity edges, while other edges based on physical proximity use a larger decay factor, $\lambda_{phys}$ (set to 0.01).

- **Physical Process Enhancement** ($\beta$)**:** Weights are further modulated by dynamic physical processes. For example, a shadow edge determined to be actively casting a shadow in the current hour has its weight multiplied by an enhancement factor, $\beta_{shadow}$ (set to 1.2).

- **Source Node Attribute Influence** ($\gamma$)**:** Weights are also influenced by the attributes of the source node. For instance, the weight of a vegetation activity edge is positively affected by the height of its source tree, $h_{tree}$, controlled by the modulation factor $\gamma_{tree}$ (set to 0.2).

Although this scheme is theoretically more physically interpretable, our ablation experiments showed no improvement in predictive performance when introducing explicit edge attributes (E) and edge weights (EW) compared to a simpler model that only uses edge types (results shown in the table A12). This suggests that, for the UrbanGraph architecture and the UTCI prediction task, the model can effectively and implicitly learn the strength of these interactions from the dynamic graph topology and node features, without needing explicitly injected edge weights and attributes.

Table A12: Ablation study on explicit edge attributes (E) and edge weights ($E_W$).

| Model | E | $E_W$ | R² | MSE |
|---|---|---|---|---|
| **Base** | | | **0.8629** | **1.0976** |
| EF | $\checkmark$ | | 0.8530 | 1.1513 |
| EFW | $\checkmark$ | $\checkmark$ | 0.8586 | 1.2097 |

### E.5 MULTI-TASK VS. SINGLE-TASK LEARNING

To evaluate whether shared representations could enhance performance, we compared the proposed Single-Task Learning (STL) framework (where a separate model is trained for each variable) against a Multi-Task Learning (MTL) variant, which uses a shared Urban-Graph encoder followed by variable-specific heads. As shown in Table A13, STL consistently outperforms MTL, with the performance drop being particularly pronounced for variables with distinct physical mechanisms (e.g., Wind Speed vs. MRT). We attribute this to negative transfer arising from the inherent heterogeneity of the underlying physical processes: fluid dynamics (governing Wind Speed) and radiative transfer (governing MRT) require learning distinct and often conflicting spatial dependencies. Forcing a shared encoder to capture these diverse physical laws dilutes the specificity of the node embeddings, confirming that independent training ensures each model specializes in the specific physical operator relevant to its target.

Table A13: Performance on target variables.

| Target | STL | MTL |
|---|---|---|
| UTCI | **0.8629 $\pm$ .0696** | 0.7460 $\pm$ .0806 |
| AT | **0.5650 $\pm$ .1324** | 0.4080 $\pm$ .1382 |
| WS | **0.7500 $\pm$ .0176** | 0.4794 $\pm$ .0315 |
| MRT | **0.8378 $\pm$ .2005** | 0.7181 $\pm$ .1700 |
| RH | **0.5159 $\pm$ .2039** | 0.4105 $\pm$ .2067 |
| PET | **0.8492 $\pm$ .0517** | 0.6883 $\pm$ .0589 |

# F  GENERALIZATION AND VISUALIZATION

## F.1  PREDICTION RESULTS ON DIFFERENT ARCHITECTURES

To provide a qualitative assessment of our model's performance, this section presents a visual comparison of the spatio-temporal prediction results between UrbanGraph and the four categories of baseline models. Each figure displays the ground truth, the predictions from UrbanGraph and representative baselines, and their respective prediction error maps (Prediction - Ground Truth) for a selected test scene at different hours of the day. White areas in the maps correspond to buildings, which are excluded from the analysis.

Figure A9 compares UrbanGraph with non-graph and homogeneous graph baselines, which represent fundamentally different approaches to spatial modeling. Figure A10 provides a comparison against generative graph models and temporal variants, assessing different graph learning strategies and sequence modeling components.

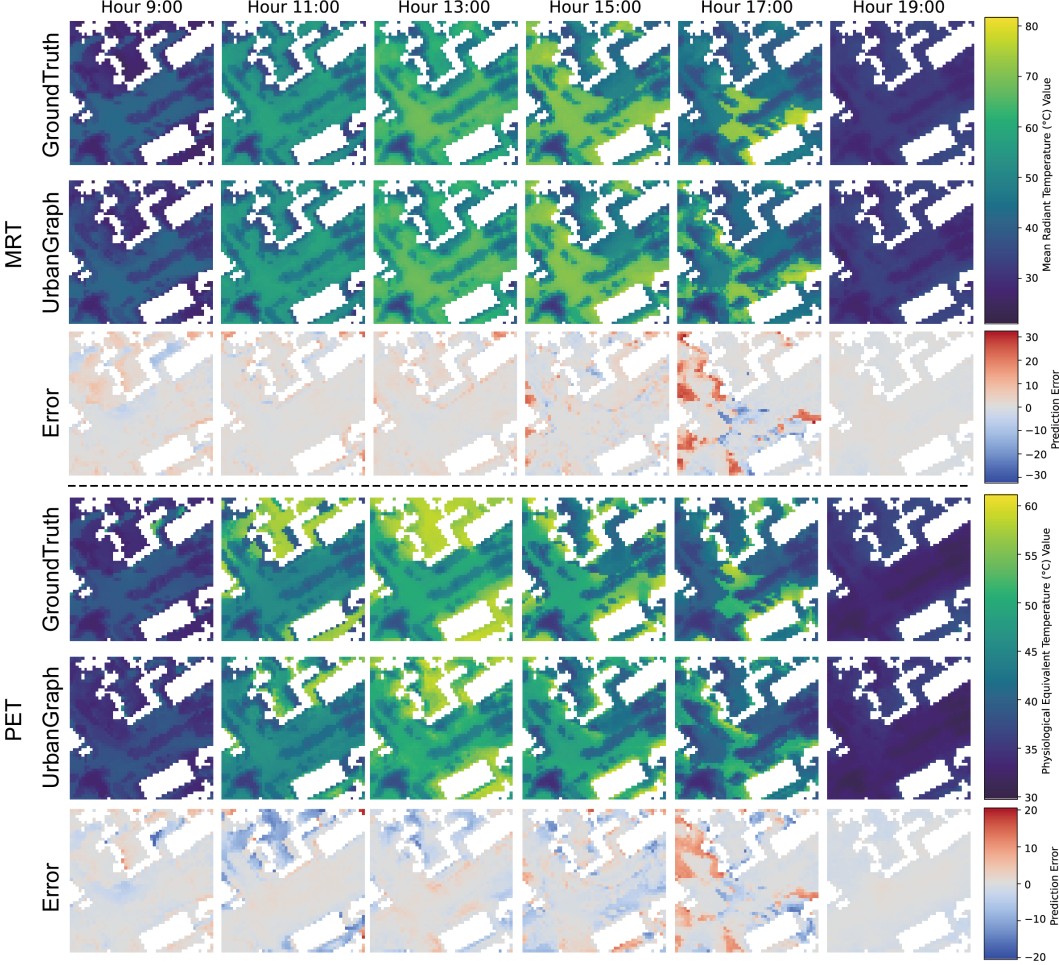

Figure A8: Qualitative prediction results for thermal comfort indices. This figure visualizes the performance of UrbanGraph on MRT and PET. Similar to the previous figure, each block compares the ground truth, model prediction, and the resulting error map, demonstrating the model's strong performance on composite indices.

## F.2  PERFORMANCE ON MULTIPLE TARGETS

The robustness of our physics-informed representation and the UrbanGraph architecture is validated by the consistent performance on five remaining target variables (Table A13), where all $R^2$ scores

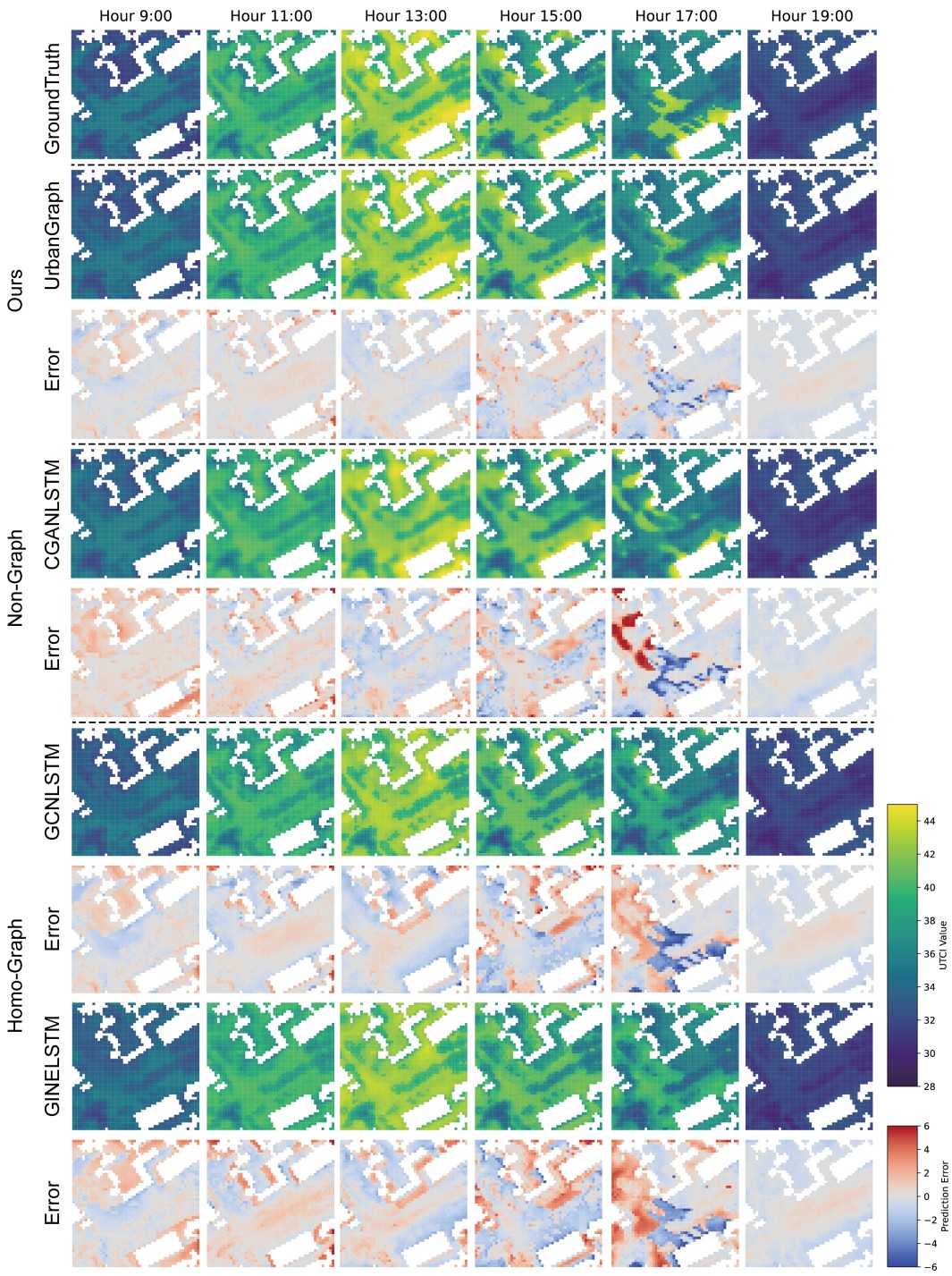

Figure A9: Visual comparison against Non-Graph (CGAN-LSTM) and Homogeneous Graph (GCN-LSTM, GINE-LSTM) baselines. Compared to the grid-based CGAN-LSTM, UrbanGraph better captures fine-grained spatial details. Unlike the homogeneous models that treat all interactions uniformly, UrbanGraph's heterogeneous approach leads to more physically consistent predictions and lower overall error.

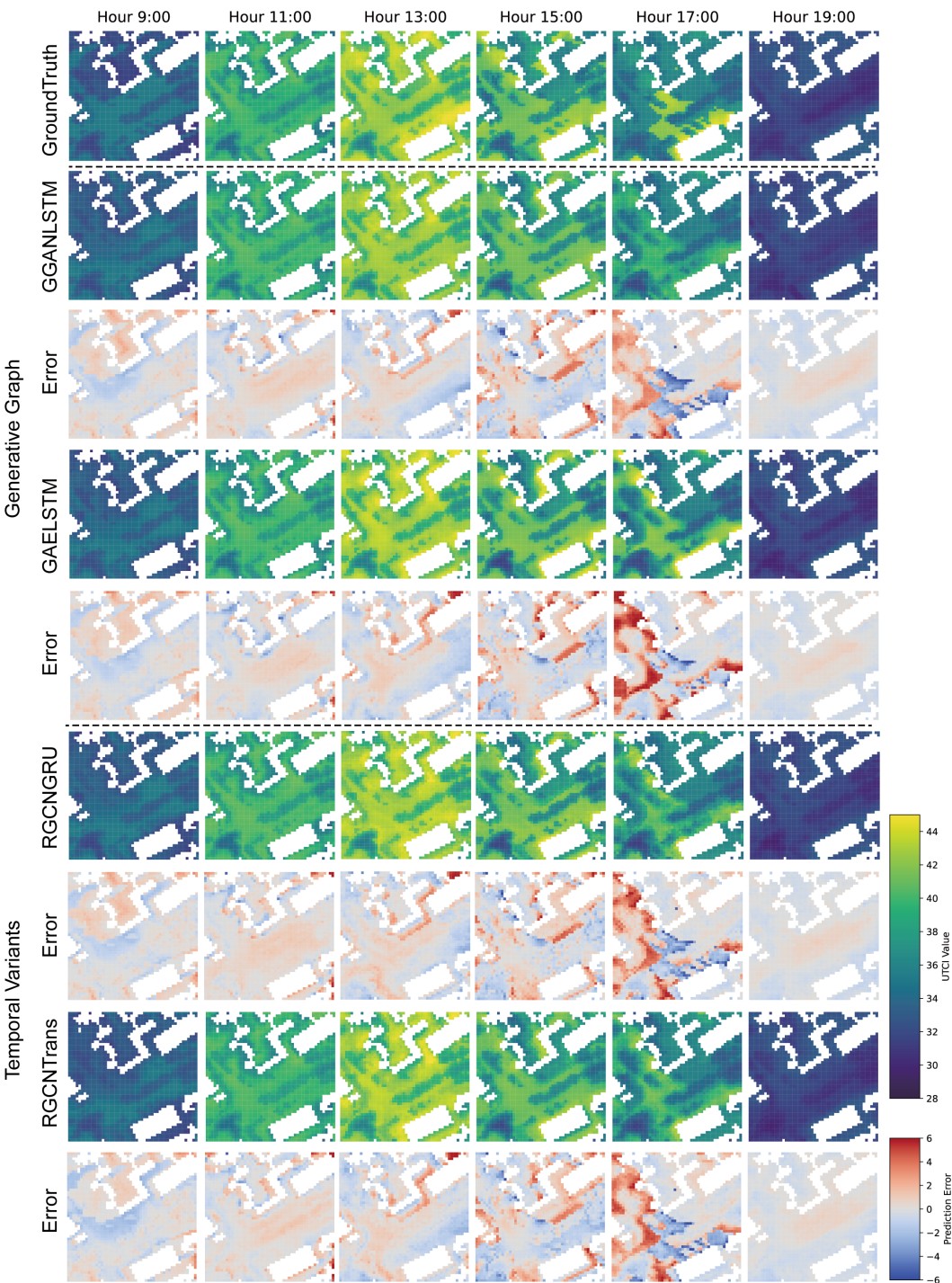

Figure A10: Visual comparison against Generative Graph (GGAN-LSTM, GAE-LSTM) and Temporal Variant (RGCN-GRU, RGCN-Transformer) baselines. UrbanGraph's physics-informed, deterministic graph construction (shown in Figure A9) avoids the higher errors seen in generative approaches. Furthermore, its LSTM component proves more effective at capturing long-term dependencies compared to the GRU and Transformer variants.

exceed $0.5$ and MRT/PET performance nearly matches UTCI. Qualitative generalization capabilities for these five variables are visualized in Figures A8– A11.

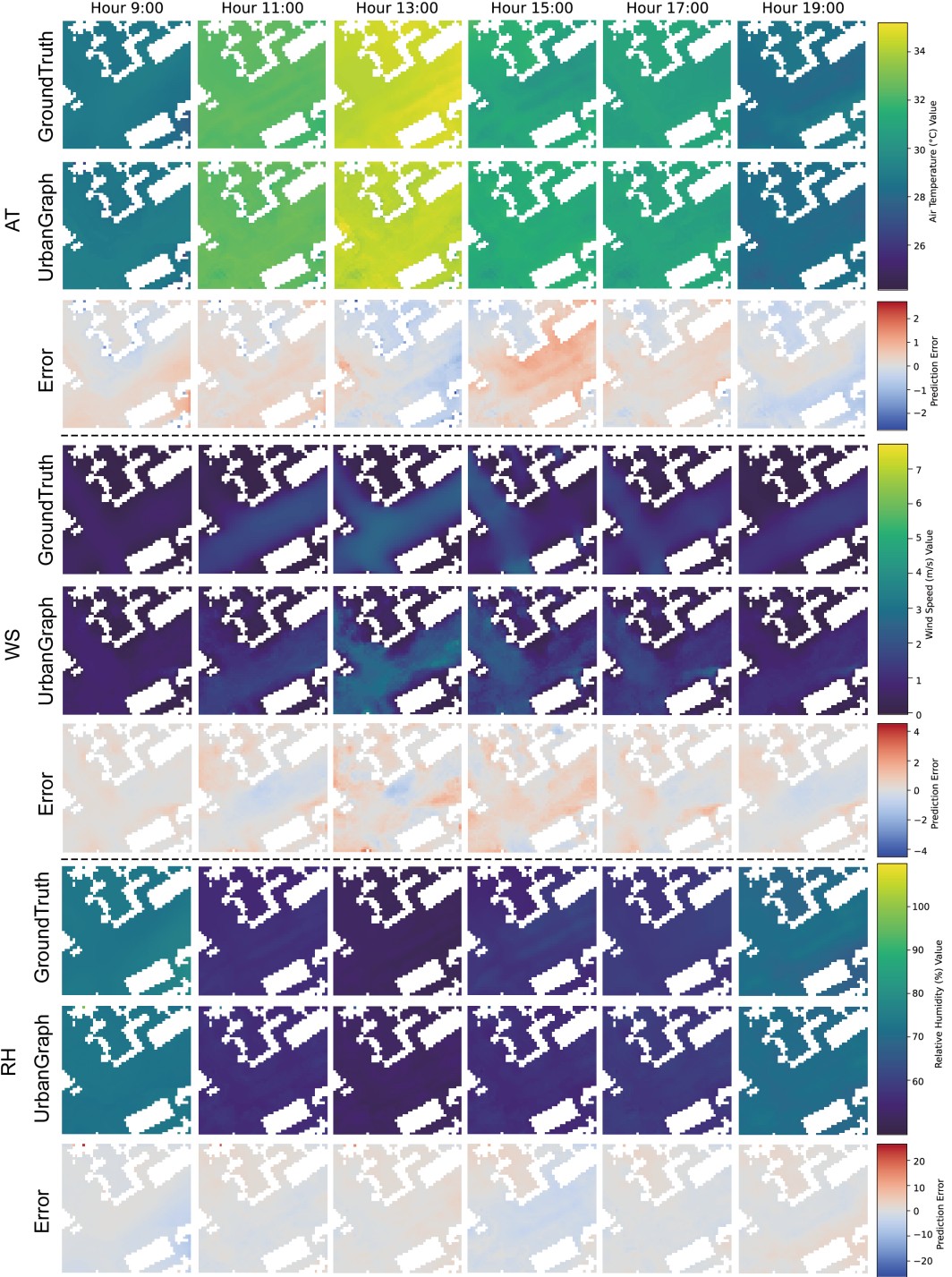

Figure A11: Qualitative prediction results for microclimate variables. This figure visualizes the performance of UrbanGraph on AT, WS, and RH. For each variable, the top row shows the ground truth, the middle row shows the model's prediction, and the bottom row displays the prediction error map across different hours.

## G  GENERALIZATION ON VECTOR FIELDS

### G.1  DATASET GENERATION AND PHYSICAL SETUP

To evaluate the model's generalization on vector fields, we utilized the UWF3D dataset. This dataset consists of high-fidelity CFD simulations generated using the open-source platform OpenFOAM, with the following specific configurations:

**Physical Modeling**. Unlike the thermal dataset generated by ENVI-met, UWF3D solves the steady-state Reynolds-Averaged Navier-Stokes (RANS) equations coupled with the standard $k - \epsilon$ turbulence model. This setup captures complex non-linear fluid dynamics such as flow separation, cavity circulation, and wake turbulence.

**Geometric Configuration**. The simulation domain represents a $540m \times 540m$ idealized urban block. Building geometries are randomly generated via parametric design (Rhino/Grasshopper) based on real-world prototypes, ensuring morphological diversity.

**Boundary Conditions**. The inflow boundary follows a logarithmic wind profile with a reference velocity of $10m/s$ at $10m$ height. The simulations operate at high Reynolds numbers ($Re \approx 10^7 - 10^8$), ensuring Reynolds independence consistent with real-world urban aerodynamics.

**Mesh and Solver**. The computational domain is discretized using an unstructured mesh containing approximately 2 to 4 million grid points per sample. This high spatial resolution allows for precise prediction of the 3D velocity vector field $\mathbf{v} = (u, v, w)$ at each spatial node.

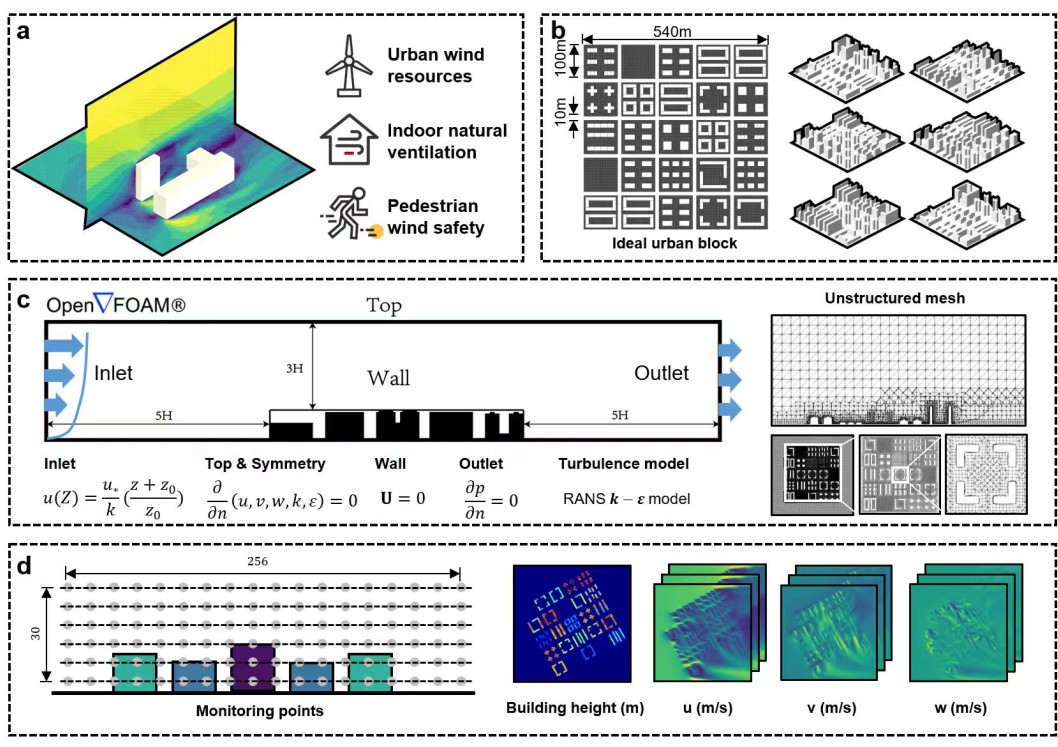

Figure A12: Visual overview of the UWF3D dataset configuration and simulation samples. The figure illustrates the diversity of parametrically generated urban geometries and their corresponding high-fidelity CFD velocity fields. The simulations, solved via RANS equations with the $k - \epsilon$ turbulence model, explicitly capture complex aerodynamic phenomena such as flow separation and wake recirculation around varying building morphologies.

### G.2 MODEL SETUP AND COMPARATIVE IMPLEMENTATION

To rigorously validate the framework's adaptability, we established a benchmark between the proposed UrbanGraph and a Grid-GCN baseline. Given that this task involves purely fluid dynamics without thermal radiation or biological processes, we adapted the graph construction rules to focus exclusively on the Navier-Stokes mechanisms.

#### G.2.1 BASELINE: GRID-GCN

The Grid-GCN serves as a control group representing the standard computer vision approach applied to physical fields.

**Graph Construction**. It constructs a homogeneous lattice graph on the $64 \times 64$ sampling grid. Connectivity is defined strictly by Euclidean proximity using the Von Neumann neighborhood (connecting only to 4 immediate neighbors: up, down, left, right).

**Mechanism**. Crucially, this topology forces isotropic message passing, where features are aggregated uniformly from all spatial directions regardless of the local wind vector. This effectively treats fluid prediction as a generic image smoothing task, ignoring the directional nature of momentum transport.

#### G.2.2 URBANGRAPH

In contrast, UrbanGraph utilizes its heterogeneous graph capabilities to encode specific fluid dynamic operators. We mapped the edge types from the main paper to this aerodynamic context:

**Advection Edges**. To simulate momentum transport, we construct directed edges searching upstream along the dominant wind vector. This explicitly models the advection process, allowing the network to transport flow features from upstream nodes to downstream ones (mimicking the semi-Lagrangian scheme).

**Wake Edges**. To capture non-local turbulence and pressure drops behind obstacles, we construct wake edges that connect building boundary nodes directly to their leeward wake zones. This is topologically analogous to the "Shading" edges in the thermal task (both representing directional obstruction effects), enforcing physical consistency in cavity circulation regions.

#### G.2.3 PERFORMANCE ANALYSIS

Quantitative results in Table A14 show that UrbanGraph achieves an $R^2$ of 0.8886 for the $u$-component, significantly outperforming the Grid-GCN. The performance gap highlights a critical physical insight: Isotropic aggregation (Grid-GCN) inevitably over-smooths the vector field, failing to maintain sharp velocity gradients at

Table A14: Performance comparison on velocity vector components ($R^2$).

| Model | $u$ | $v$ | $w$ |
|---|---|---|---|
| Grid-GCN | $0.7827 \pm 0.0100$ | $0.6885 \pm 0.0166$ | $0.6858 \pm 0.0141$ |
| **UrbanGraph** | $\mathbf{0.8886 \pm 0.0004}$ | $\mathbf{0.8474 \pm 0.0025}$ | $\mathbf{0.7937 \pm 0.0020}$ |

separation points. Conversely, UrbanGraph's anisotropic inductive bias successfully reconstructs complex flow patterns, such as flow separation and wake recirculation, by explicitly routing information along physically valid flow paths.

