# OpenReview forum: "UrbanGraph: Physics-Informed Spatio-Temporal Dynamic Heterogeneous Graphs for Urban Microclimate Prediction"
_ICLR.cc/2026/Conference — ICLR 2026 Poster_

### Official Review · Reviewer_mEEd · 2025-10-27

**Soundness:** 3
**Presentation:** 4
**Contribution:** 3
**Rating:** 6
**Confidence:** 4

**Summary:**

The authors propose a novel method to generate high resolution forecast of weather variables over cities. The method employs a physics informed graph representation that explicitly maps physical processes.
The authors propose to discretize GIS features and then explicitly express the relationship between cells in five different edge types: Vegetation Evapotranspiration, Shading, Convective Diffusion, Semantic Similarity, Internal Contiguity.
These different edge types govern how the different cells are connected to each-other.

The proposed method is trained on and compared to a high resolution numerical model output (ENVI-met), and performs better than several other baseline methods.

**Strengths:**

The paper is well written, the advantages of the methods clear and the figures of very high quality. The arguments for the design choices are sound, and make sense in the context of the paper.
The five types of edges seem to accurately represent the different processes, and suffice to express the complex relationships between nodes.
The model does seem to perform better than a series of baselines both in accuracy and time cost.

**Weaknesses:**

Although the paper is quite strong, I feel like a few key weaknesses exist:
- The authors claim that previous methods mostly use fixed graph structures for time series forecasting. Although this might be true, I would argue that their method is also fixed. The graph is a gridded mesh, where the edges weight are varying. Additionally, the advantage of using GNNs is usually to work with non-gridded data. Since the domain is discretized, what is the advantage of using a GNN? Wouldn't a transformer with a weighted attention mask work as well?
- There is no mention of spatial resolution. Maybe I've missed it, but what is the resolution? The GIS data is discretized, so there is a spatial resolution.
- I am not sure I understand what the inputs are. Beyond the discretized GIS data, what are the inputs of the model? How are the environmental variables integrated? I also didn't see any mention of embedding the environmental variables. The only mention is the Graph-level global environmental features. Are the environmental features only embedded at the graph level? Is there no per-pixel data?
- I have only seen a train a validation set mentioned, but not test set.
- The authors claim to increase the computational efficiency in FLOPS by 17%. But in table 1, GGAN-LSTM performs better in terms of FLOPS. Where does this come from?
- In table 1, URBANGRAPH is bold for most results, but several models beat it for time cost.


Minor weaknesses:
- Figure 4:
	- a) it is very uncommon to show train and validation results. Results are usually shown on the test set.
	- a) What is the x axis?
	- b) the colors are hard to differenciate
	- b) where is URBANGRAPH?
- Figure 3: I don't understand what the discretized images are (top left)
- Line 256: "normalized static feature space." which normalized static feature space?
- Line 379: MAE is not reported

**Questions:**

I am not sure I understand how the final network is constructed, can i get an example of network?
I would like to see examples of results in the main paper.

---

> ### Author Response · Authors · 2025-11-21
> **Response to Reviewer mEEd's Comments**
>
> We sincerely thank you for your thorough assessment. We have implemented comprehensive revisions to address every major and minor point, significantly enhancing the clarity and rigor of the manuscript.
>
> Our point-by-point responses are attached below. All revisions in the manuscript are marked in blue.
>
> $\textbf{Response to W1:}$
>
> We highly appreciate your critical and deep questions regarding the use of GNNs in a discrete domain, which address the core trade-off of our methodology.
>
> 1.	While nodes are fixed on a grid, the graph connectivity is structurally dynamic (edges are created/deleted based on instantaneous physics, e.g., shading), forming a time-varying sparse subgraph—not merely a re-weighted fixed graph.
> 2.	GNNs are essential because urban physical interactions are non-Euclidean, non-local, and anisotropic (e.g., long-range shading, directional wind). GNN edges directly and efficiently model these complex physical causal pathways.
> 3.	Although Transformer is a valid alternative, our method provides a Hard Structural Inductive Bias based on physics, which is more data-efficient and empirically proves to yield superior performance ($R^2=0.8542$ vs $0.8465$) for this prediction task.
>
> We have revised Section 1, 4.2 and 6.2 to clarify the dynamic topology and GNN advantages.
>
> $\textbf{Response to W2:}$
>
> We apologize for the oversight regarding the spatial resolution. We confirm the primary UMC4/12 dataset uses a 4-meter horizontal resolution (3-meter vertical). For the newly added UWF3D fluid dynamics experiment (Appendix G), we utilized an even finer 2.1-meter horizontal resolution.
>
> We have updated Section 5 and Appendix B.1/G.1 for clarification.
>
> $\textbf{Response to W3:}$
>
> We highly appreciate this specific question which improves the clarity of our methodology.
>
> 1.	The model uses Static GIS Features (Building Height, Tree Height, Land Cover ID; which are node-level/pixel-wise), Dynamic Global Features (e.g., Solar Radiation, Wind Speed/Direction, Background Temperature; which are graph-level forcing data), and Time Features. High-resolution weather maps are the prediction goal, not the input.
> 2.	Global environmental variables are integrated via Feature Fusion (broadcasted to all nodes) and Topological Encoding (explicitly dictating the dynamic creation and direction of physical edges).
>
> We have updated Section 3 and Figure 4 for clarification.
>
> $\textbf{Response to W4:}$
>
> We thank the reviewer for pointing out this lack of clarity. We confirm that the entire dataset was spatially partitioned into Training (70%), Validation (20%), and Test Set (10%).
>
> We have revised Section 5 to explicitly state this splitting protocol.
>
> $\textbf{Response to W5:}$
>
> We thank you for the query regarding computational efficiency. The initial 17% FLOPS claim was the maximum gain over prior benchmarks. Our advantage is now clear: UrbanGraph ($R^2=\mathbf{0.8542}$) reduces FLOPS by $\approx$ 73.8% compared to LRGCN ($3.49 \times 10^{10}$), validating the high efficiency of our physics-constrained structure.
>
> We have revised Section 6.1.
>
> $\textbf{Response to W6:}$
>
> We thank you for your comment regarding time cost. Faster models (e.g., STGCN) gain speed by relying on static/isotropic structures, but sacrifice substantial accuracy ($R^2=0.7880$ vs $\mathbf{0.8542}$) because they fail to model dynamic physical interactions. UrbanGraph's core advantage is securing the best trade-off between the highest accuracy and computational cost.
>
> We have revised Section 6.1.
>
> $\textbf{Response to Minor W1:}$
>
> We thank the reviewer for pointing out the visualization issues. We have completely redrawn Figure 5 (formerly Figure 4), clarifying the x-axis ("Training Epochs"), updating the color scheme, and explicitly highlighting UrbanGraph to meet standard presentation norms.
>
> We have updated Figure 5 and its caption accordingly.
>
> $\textbf{Response to Minor W2:}$
>
> We apologize for the unclear labeling. The image in Figure 3 (top-left) represents the Static GIS Features (building height, Tree Height, and land cover type) being fed into the network's initial embedding layers.
>
> We have updated the Figure 3 caption accordingly.
>
> $\textbf{Response to Minor W3:}$
>
> We thank the reviewer for requesting this clarification. The "normalized static feature space" refers specifically to the three static GIS features (Building Height, Tree Height, and Land Cover Type) after being Z-Score normalized to prevent features with larger magnitudes from dominating the k-NN calculation.
>
> We have updated Section 4.1 accordingly.
>
> $\textbf{Response to Minor W4:}$
>
> We apologize for the omission. We have added the MAE metric for all models to Table 1 in the revised manuscript.
>
> $\textbf{Response to Q1:}$
>
> We thank the reviewer for this suggestion. We have added a spatial heatmap visualization (Figure 4) to Section 6.1 that visually presents the key input data alongside the comparison between the ENVI-met Ground Truth and the UrbanGraph prediction.

---

> > ### Author Response · Authors · 2025-11-27
> >
> > Dear Reviewer mEEd,
> >
> > We sincerely thank you for your constructive feedback, which has been incredibly helpful in refining our work. We have carefully updated the paper with additional experiments and provided a detailed point-by-point response to address your specific comments.
> >
> > As the discussion period is coming to an end, we would appreciate it if you could let us know if our rebuttal has satisfactorily resolved your concerns. We remain available to answer any further questions you might have.
> >
> > Thank you again for your time and effort.
> >
> > Best regards,
> > The Authors

---

### Official Review · Reviewer_dxkG · 2025-10-29

**Soundness:** 3
**Presentation:** 3
**Contribution:** 3
**Rating:** 6
**Confidence:** 3

**Summary:**

This paper presents UrbanGraph, a physics-informed framework for urban microclimate prediction using dynamic heterogeneous graphs. The authors propose encoding multiple physical processes (shading, vegetation evapotranspiration, convective diffusion) as different edge types in a heterogeneous graph structure, where edges are dynamically reconstructed at each timestep based on environmental conditions. The framework combines Relational Graph Convolutional Networks (RGCN) for spatial feature extraction with LSTM for temporal modeling. The authors evaluate their approach on UMC4/12, a new dataset containing 11.9 million high-resolution spatio-temporal data points generated from ENVI-met simulations across 11 urban sites in Singapore. The results show improvements of up to 10.8% in R² and 17.0% reduction in FLOPs compared to baselines.

**Strengths:**

1、Well-motivated physics-informed design:
The explicit encoding of physical processes into graph topology is intuitive and grounded in urban climate science. The distinction between static edges (semantic similarity, internal contiguity) and dynamic edges (shading, vegetation activity, convective diffusion) appropriately captures both time-invariant spatial relationships and temporally evolving physical phenomena.
2、Dataset contribution:
The UMC4/12 dataset with diverse urban morphologies and high-resolution simulations provides a valuable resource for the research community.
3、Clear presentation:
The paper is generally well-written with good use of figures to illustrate the framework and results.

**Weaknesses:**

the explicit encoding of physical processes (such as shading, vegetation evapotranspiration, and convective diffusion) into a dynamic, heterogeneous graph topology is a novel and effective method. however, this heavy reliance on predefined rules raises critical questions about the model's flexibility and potential for scientific discovery.  The model is constrained by the five predefined relationship types. As the authors correctly acknowledge in the limitations section, this may oversimplify the real, complex physical interactions.

**Questions:**

1 The ablation studies and baseline comparisons effectively demonstrate the superiority of UrbanGraph over simpler models like standard GCNs (proving the value of heterogeneity) and non-graph models like CGANs. However, the paper is missing a comparison against more advanced, integrated spatio-temporal Graph Neural Networks (e.g., STGCN, Graph WaveNet, ASTGCN, or ST-GNNs designed for traffic). These models often feature more sophisticated spatio-temporal fusion mechanisms than the sequential RGCN-then-LSTM approach used here. Could the authors comment on why these state-of-the-art spatio-temporal GNNs were not included as baselines? A discussion (or ideally, a new experimental comparison) would be needed to truly position UrbanGraph's performance within the broader ST-GNN literature, rather than just against its own ablated components.
2 Some parameters are likely effective for the UMC4/12 dataset, which is based on Singapore's climate. a) How sensitive is the model's accuracy to these specific values? b) If this model were deployed in a city with a vastly different climate and morphology (e.g., a dry, arid climate with sparse vegetation or different wind patterns), would these parameters need to be manually re-tuned? c) Does the model risk failing if the true physical influence (e.g., of wind) differs from the hard-coded heuristic?

---

> ### Author Response · Authors · 2025-11-21
> **Response to Reviewer dxkG's Comments**
>
> Thank you for your professional and invaluable feedback. We fully accepted your key suggestions regarding the comparison with advanced baselines and parameter sensitivity/transferability. We have thoroughly revised the manuscript with new experiments and detailed discussions, significantly enhancing the paper's rigor. We appreciate your crucial contribution.
>
> Our point-by-point responses are attached below. All revisions in the manuscript are marked in blue.
>
> $\textbf{Response to W1:}$
>
> We sincerely thank you for this critical and highly valuable comments. We acknowledge that predefined rules introduce a strong inductive bias. However, unlike de novo exploration tasks such as protein generation, urban microclimate is governed by established physical laws. Our goal is to apply these laws for rapid, reliable engineering prediction. Explicitly encoding the dominant physical processes provides the necessary structural inductive bias that ensures engineering reliability and consistency. Crucially, flexibility is maintained because the RGCN learns distinct weights ($W_r$) for each relation, allowing data-driven corrections that mitigate the oversimplification introduced by the physical heuristics.
>
> We view this as the foundation for reliable modeling. To achieve greater flexibility and scientific discovery potential, we propose a future Hybrid Topology Strategy to automatically discover and encode unmodeled complex interactions while preserving physical consistency.
>
> We have updated Section 7 (Limitations and Future Work).
>
> $\textbf{Response to Q1:}$
>
> We sincerely appreciate your suggestion to contextualize UrbanGraph within the broader ST-GNN literature, which we agree is essential. To address this, we have significantly expanded our evaluation by incorporating four categories of SOTA baselines: $\textbf{1) Grid-based}$(CGAN-LSTM, Pix2Pix+PINN), $\textbf{2) Static ST-GNNs}$ (STGCN, ASTGCN), $\textbf{3) Generative Graph models}$ (GGAN-LSTM, GAE-LSTM), and $\textbf{4) Dynamic Graph models}$ (LRGCN).
>
> The updated results (Table 1) consistently confirm UrbanGraph's superior empirical performance ($R^2=0.8542$) over all advanced baselines, including the strongest integrated competitor, LRGCN ($R^2=0.8422$). Furthermore, comparison with PINN and generative models highlights that our hard structural constraint (via topology) is more effective for ensuring physical consistency than soft loss constraints or implicit structure learning.
>
> We assert that our sequential "RGCN-then-LSTM" decoupled architecture is theoretically optimal for this domain. This Divide-and-Conquer approach uses explicit physical rules to resolve dynamic topology first, providing the LSTM with a clean, physics-consistent state. This focus on Markovian temporal evolution (which aligns better with urban physics PDEs) results in a 21% faster training time compared to the integrated LRGCN.
>
> We have updated Section 5 (Baselines), Section 6.1 (Performance Analysis), and Appendix E.1.
>
> $\textbf{Response to Q2a:}$
>
> We appreciate this insightful question. The parameters (e.g., $R_{max}^{shadow}$) are essential physics-based priors derived from robust urban physics literature, capturing fundamental physical scales. A new UWF3D experiment confirms the empirical robustness of these priors, achieving high accuracy ($R^2 > 0.8886$) on an independent dataset without fine-tuning. This validates they are not overfitted.
>
> We have revised Section 4.1, Appendix C, and Appendix G for clarification.
>
> $\textbf{Response to Q2b:}$
>
> We thank you for this crucial question on transferability. Manual retuning is generally not required, but it is necessary if the underlying physical regime fundamentally changes. Edge attributes are derived from first-principle equations (Eqs. 3-7), allowing the graph to automatically adapt to input changes (e.g., humidity, radiation), handling most variations natively. Validation across Singapore's diverse micro-climates (Appendix B.2) confirms robustness for typical urban variations. We acknowledge that for extreme regimes (where new physical mechanisms dominate), adjusting the empirical priors is needed.
>
> We have revised Section 6.1 and Appendix B.2 for clarification.
>
> $\textbf{Response to Q2c:}$
>
> We thank you for identifying this crucial risk. We acknowledge that simplified heuristics may not fully capture complex physics (e.g., turbulence). However, our framework mitigates failure through two mechanisms: 1) Soft Correction: The RGCN's learnable weights ($W_r$) adaptively suppress inaccurate heuristic connections. 2) Structural Redundancy: The inclusion of data-driven Semantic Similarity Edges (k-NN) creates a redundant information pathway, maintaining performance when heuristics fail locally. We propose the Hybrid Topology approach (adding a learnable Residual Graph) as future work to automatically discover and compensate for the remaining gaps.
>
> We have added a discussion on this fault tolerance mechanism to Section 4.1.

---

> > ### Author Response · Authors · 2025-11-27
> >
> > Dear Reviewer dxkG,
> >
> > We sincerely thank you for your constructive feedback, which has been incredibly helpful in refining our work. We have carefully updated the paper with additional experiments and provided a detailed point-by-point response to address your specific comments.
> >
> > As the discussion period is coming to an end, we would appreciate it if you could let us know if our rebuttal has satisfactorily resolved your concerns. We remain available to answer any further questions you might have.
> >
> > Thank you again for your time and effort.
> >
> > Best regards,
> > The Authors

---

### Official Review · Reviewer_jFx8 · 2025-10-31

**Soundness:** 3
**Presentation:** 4
**Contribution:** 3
**Rating:** 6
**Confidence:** 3

**Summary:**

This paper proposes UrbanGraph, a framework for urban microclimate prediction that integrates physics-informed, dynamic, and heterogeneous graph neural networks. The key idea is to explicitly encode time-varying physical processes, such as shading, vegetation evapotranspiration, and convective diffusion, into the topology of a dynamic heterogeneous graph. This graph structure is then processed by a spatio-temporal model combining a Relational Graph Convolutional Network (RGCN) for spatial dependencies and an LSTM for temporal evolution. Moreover, the authors also curate the UMC4/12 dataset, a high-resolution, physics-based simulation benchmark. Empirical experiments demonstrate that UrbanGraph outperforms various strong baselines.

**Strengths:**

1. The problem is well-motivated.
2. The proposed idea of encoding observational/structural bias directly into the topology, i.e., physics-informed edges rather than physics-informed losses, is principled and technically sound.
3. The dataset contribution is valuable to the community.
4. The paper is well written.

**Weaknesses:**

1. The method is evaluated on a CFD-style simulator (ENVI-met) under several urban configurations. This is fine for an anonymized submission, but the key claim is physics-informed generalization to realistic urban microclimates. Without any real-world/field-sensor validation, it’s hard to tell if the hand-crafted dynamic edge rules are robust to noisy or incomplete inputs. Would it be possible to deploy on real city data (even small-scale)?
2. The model uses an LSTM as the temporal evolution module. They compare to GRU and Transformer, but the argument for LSTM as the final choice is mostly empirical. Given that the graph itself is dynamic, a temporal attention or cross-time graph operation might better exploit the structured changes in edges.

**Questions:**

1. How expensive is rebuilding 3 physics-driven edge sets every hour for a large city-level grid (say 50k–100k nodes)? Or have you considered or tested any strategies to manage this complexity?
2. You report on six target variables (UTCI, PET, AT, MRT, WS, RH). Are they trained with a single shared UrbanGraph and separate heads, or trained separately per target (Eq. (1) suggests separate mappings)? If separate, could multitask training further boost R² through shared spatial embeddings?

---

> ### Author Response · Authors · 2025-11-21
> **Response to Reviewer jFx8's Comments**
>
> We thank the reviewer for recognizing the novelty of encoding structural biases directly into the topology. Your insightful questions motivated us to strengthen our empirical validation. In response, we incorporated two-tier real-world validation (sensor/city-scale), verified large-scale scalability via stress tests, and conducted comparative experiments on multi-task learning. We hope these revisions address your concerns.
>
> Our point-by-point responses are attached below. All revisions in the manuscript are marked in blue.
>
> $\textbf{Response to W1:}$
>
> We thank the reviewer for highlighting the importance of real-world validation. To verify the robustness of our physical rules against real-world noise, we implemented a two-tier validation strategy:
>
> $\textbf{1) Micro-scale Calibration}$:
>
> We rigorously validated our simulation engine (ENVI-met) against in-situ NUS sensor measurements. The strong correlation ($r > 0.73$) confirms the physical fidelity of our training data source.
>
> $\textbf{2) City-scale Generalization}$:
>
> We deployed UrbanGraph to predict thermal stress across all of Singapore. High agreement ($r = 0.842$) with the independent HiGTS dataset demonstrates the model's effectiveness in complex, real-world scenarios.
>
> We have added Appendix B.2 to detail these real-world validation experiments.
>
> $\textbf{Response to W2:}$
>
> We thank the reviewer for this insightful suggestion. While our initial choice appeared empirical, we conducted extensive comparative experiments (incorporating ASTGCN, RGCN-Transformer, and LRGCN) to rigorously verify this design. The results confirm that the Decoupled LSTM architecture is superior for this specific physical task due to two theoretical alignments:
>
> $\textbf{1) Alignment with Physical Markovianity (vs. Attention)}$:
>
> UrbanGraph ($\text{R}^2=0.8542$) outperforms attention-based models (e.g., ASTGCN $\text{R}^2=0.8317$) and RGCN-Transformer ($\text{R}^2=0.8465$). This superiority stems from LSTM's recurrent structure naturally aligning with the Markovian nature of microclimate physics, where the future state evolves continuously from the immediate past. Global attention mechanisms, in contrast, are less critical for continuous thermodynamic evolution.
>
> $\textbf{2) Alignment with Spatial-Temporal Decoupling (vs. Cross-time Ops)}$:
>
> UrbanGraph outperforms the integrated LRGCN ($\text{R}^2=0.8422$) and is 21% faster in training. This validates our "Divide-and-Conquer" strategy: by explicitly resolving the dynamic topology via the physics-informed RGCN before the temporal stage, the decoupling strategy avoids the optimization difficulty of simultaneously learning structural evolution and temporal dynamics ("entanglement"), offering a more efficient solution than integrated cross-time operations.
>
> We have updated Section 4.2 to articulate this design rationale regarding Markovian alignment and decoupling.
>
> $\textbf{Response to Q1:}$
>
> We thank the reviewer for raising this critical question regarding scalability. To directly address the concern about handling 50k–100k nodes, we conducted an empirical stress test on a large urban region comprising approximately 90,000 spatial nodes (partitioned into 36 overlapping windows). The total time required to reconstruct all physics-based edges for the entire 13-hour sequence was merely 39.56 seconds (using a 16-core CPU). This empirical evidence, supported by the linear cost correlation analyzed in Appendix E.2 (Figure A7), conclusively demonstrates that our divide-and-conquer strategy effectively ensures scalability, making the edge reconstruction cost computationally negligible (< 1 minute) compared to the hours required for numerical simulations.
>
> We have added these scalability results to Appendix E.2.
>
> $\textbf{Response to Q2:}$
>
> We appreciate the reviewer's valuable suggestion to explore parameter sharing for potential performance gains. We confirm that the original submission used Single-Task Learning (STL). Motivated by your comment, we implemented a Multi-Task Learning (MTL) variant where a shared UrbanGraph encoder extracts unified node embeddings for all six variables.
> Contrary to standard expectations, STL consistently outperforms MTL (e.g., UTCI $R^2$: 0.8629 vs. 0.7460). We attribute this to negative transfer arising from physical heterogeneity:
>
> $\textbf{1) Conflicting Physical Laws}$:
>
> Target variables are governed by fundamentally different equations. For instance, Wind Speed follows fluid dynamics (Navier-Stokes, dependent on flow connectivity), whereas MRT follows radiative transfer (dependent on line-of-sight shading).
>
> $\textbf{2) Task Interference}$:
>
> Forcing a single encoder to capture these conflicting spatial dependencies (e.g., vector anisotropy vs. scalar diffusion) dilutes the specificity of the node embeddings. Thus, independent training ensures that each model learns the specialized "physical operator" optimal for its specific target.
>
> We have added these comparison results to Appendix E.5.

---

> > ### Comment · Reviewer_jFx8 · 2025-11-26
> >
> > Thank you for the rebuttal and additional results. In short, most of my concerns have been addressed, and I will keep my positive score.

---

> > > ### Author Response · Authors · 2025-11-27
> > >
> > > Dear Reviewer jFx8,
> > >
> > > Thank you for your feedback. We are glad to hear that our rebuttal has successfully addressed your concerns. We sincerely appreciate your time, constructive comments, and your support for our work.
> > >
> > > Best regards,
> > > The Authors

---

### Official Review · Reviewer_nd8D · 2025-11-01

**Soundness:** 3
**Presentation:** 3
**Contribution:** 2
**Rating:** 4
**Confidence:** 4

**Summary:**

The paper introduces UrbanGraph, a physics-informed framework to predict urban microclimate. UrbanGraph integrates five physical representations including veg. evapotranspiration, shading, diffusion, similarity and internal continuity and encodes them using a relational GCN. Then, the spatial and temporal contexts are modeled using MLPs and LSTM, along with the physical constraints to predict future urban microclimate. Experiments are conducted using the simulation dataset from ENVI-met, and seven baselines are used to demonstrate the improved effectiveness and efficiency of the proposed model.

**Strengths:**

S1: High-resolution urban simulations and modeling is an interesting and timely topic for digital twins.

S2: Examples are well prepared and help present the main idea and workflow.

S3: Experiments are conducted using 7 different methods showing improvements.

**Weaknesses:**

W1: The physics-informed contribution uses different physical representations like shading. This is reasonable but the methodological novelty is not well explained. It seems more to be an application that uses basic physics knowledge to define the graph and the physics is specific to this problem.

W2. The experiments feel like an ablation study. More standalone and recent methods should be included for comparison. It uses only a single dataset or comparison. Using additional datasets can help evaluate the generalizability of the model.

W3: In the appendix, the tree and building layers are the same for the second row in Figure A1. This raises concerns about data correctness.

**Questions:**

It will be helpful if the authors can clarify what the technical contributions are for the edge designs in Figure 1.

---

> ### Author Response · Authors · 2025-11-21
> **Response to Reviewer nd8D's Comments**
>
> We sincerely thank the reviewer for their critical and constructive feedback, and for recognizing the timely nature of our work. We fully acknowledge the necessity to clarify methodological novelty and strengthen broader benchmarking. Your insightful questions served as an instrumental guide, prompting us to significantly elevate the theoretical justification of our edge design and expand the empirical validation. We hope these revisions address your concerns.
>
> Our point-by-point responses are attached below. All revisions in the manuscript are marked in blue.
>
> $\textbf{Response\ to\ W1:}$
>
> We sincerely thank the reviewer for this critical and inspiring observation. Your comment has prompted us to significantly elevate the theoretical positioning of our work, clarifying that UrbanGraph establishes a Structure-based Inductive Bias tailored for urban physics. We have refined our methodological contributions into four key architectural innovations:
> 1) $\textbf{Dynamic Causal Pruning}$: We explicitly frame the time-varying topology as a hard structural constraint. This forces the model's receptive field to rigorously align with the changing physical domain of influence, solving the challenge of modeling time-varying causality.
> 2) $\textbf{Physical Operator Decoupling}$: The heterogeneous design functions as a process-disentanglement mechanism. This allows the network to learn specialized operators for distinct governing equations (e.g., radiative transfer vs. fluid convection), resolving causal entanglement.
> 3) $\textbf{Fluid-Dynamic Anisotropic Modeling}$: The convective edge introduces a wind-modulated anisotropic metric into the topology. This enables the approximation of direction-dependent advection processes without the computational cost of mesh-based CFD simulations.
> 4) $\textbf{Alignment with Urban Thermodynamics}$: To address the domain uniqueness, our architecture matches the Markovian nature of thermodynamic evolution. By resolving spatial fluxes (RGCN) before time-stepping (LSTM), this spatial-temporal decoupling mimics numerical solvers, offering a more theoretically grounded approach.
>
> We have revised the Abstract, Section 1, Section 2, Section 4.1, Section 4.2, Section 6.2, and Section 7.
>
> $\textbf{Response to W2:}$
>
> We sincerely thank the reviewer for this constructive criticism. Motivated by your suggestion, we have significantly expanded our experimental scope to include four independent baselines and a completely new physical dataset to demonstrate robustness.
>
> $\textbf{1. Comparison with Independent SOTA Methods}$
>
> To address the critique that previous experiments resembled ablation studies, we benchmarked against four independent SOTA models, including LRGCN (Dynamic Graphs), STGCN/ASTGCN (Static Graphs), and Pix2Pix+PINN (Physics-Informed Generative Model). UrbanGraph consistently outperforms these advanced baselines ($R^2=0.8542$). Notably, it surpasses the strongest dynamic competitor, LRGCN ($0.8422$), while being 21% faster in training. This validates that our explicit causal pruning strategy is more computationally efficient than the implicit recurrent structure learning. Furthermore, the superiority over Pix2Pix+PINN ($0.8320$) confirms that integrating physics as a hard structural constraint ensures better consistency than soft loss constraints.
>
> $\textbf{2. Generalization on New Dataset: UWF3D}$
>
> For rigorous generalization evaluation, we constructed UWF3D, a high-resolution dataset recording vector fields governed by Navier-Stokes equations. UrbanGraph successfully generalizes to this distinct physical domain, achieving high accuracy ($u$-$R^2$: 0.8886) and significantly outperforming the Grid-GCN baseline. This strong performance across distinct physical domains (thermal diffusion vs. fluid dynamics) confirms the robustness of our explicit topological encoding framework.
>
> We have updated Section 5 and Section 6.1 (Table 1), and added Appendix E.1 and Appendix G to incorporate these results.
>
> $\textbf{Response to W3:}$
>
> We thank the reviewer for this meticulous observation. We apologize for the confusion and confirm the duplication in Figure A1 was strictly a visualization artifact (plotting script error), not an issue with the underlying data tensors. Data integrity is verified by our ablation study (Table A11): removing vegetation-specific edges caused a distinct performance drop ($R^2: 0.8629 \to 0.8504$), confirming the model successfully distinguishes between tree and building inputs.
>
> We have corrected Figure A1 in the revised manuscript.
>
> $\textbf{Response to Q1:}$
>
> We sincerely thank the reviewer for this helpful suggestion. Figure 1(b) is not merely a schematic but the visual implementation of our Structure-based Inductive Bias: time-varying connections represent Dynamic Causal Pruning, while distinct edge types enable Physical Operator Decoupling.
>
> We have updated the Figure 1(b) to explicitly label these architectural contributions.

---

> > ### Author Response · Authors · 2025-11-27
> >
> > Dear Reviewer nd8D,
> >
> > We sincerely thank you for your constructive feedback, which has been incredibly helpful in refining our work. We have carefully updated the paper with additional experiments and provided a detailed point-by-point response to address your specific comments.
> >
> > As the discussion period is coming to an end, we would appreciate it if you could let us know if our rebuttal has satisfactorily resolved your concerns. We remain available to answer any further questions you might have.
> >
> > Thank you again for your time and effort.
> >
> > Best regards,
> > The Authors

---

### Author Response · Authors · 2025-12-02
**Update on Manuscript Version for Paper 17804**

Dear Area Chair,

We have replaced the previous version (which had changes marked) with a clean version without revision marks. If you or the reviewers require the version with changes marked for reference, please let us know, and we will be happy to provide it immediately.

Best regards,
The Authors

---

### Meta-Review · Area_Chair_9jcj · 2026-01-05

**Summary:**

The reviewers expressed a consensus on the novelty and significance of the UrbanGraph framework, particularly highlighting its structure-based inductive bias and the contribution of the high-resolution UMC4/12 benchmark dataset. The authors provided a persuasive rebuttal by incorporating advanced baselines (e.g., LRGCN, Pix2Pix+PINN), demonstrating real-world validity and scalability, and verifying transferability to a new vector-field prediction task (UWF3D). Most of the reviewers give a borderline accept decision.

**Reviewer Concerns:**

Addressed Concerns:

1. **Comparison with State-of-the-Art Baselines (Reviewers nd8D, dxkG):**
The initial concern that the experiments lacked comparisons with advanced Spatio-Temporal GNNs was addressed. The authors added four categories of baselines, including LRGCN (dynamic), STGCN/ASTGCN (static), and Pix2Pix+PINN.
2. **Generalization and Real-World Validity (Reviewers jFx8, nd8D):**
Concerns regarding the reliance on simulation data (ENVI-met) and the potential "sim-to-real" gap were addressed. The authors introduced a two-tier real-world validation strategy, showing strong correlations against in-situ sensors and city-scale datasets. Furthermore, the vector-field prediction task transfer using UWF3D demonstrated the model's robustness beyond the training domain.
3. **Scalability of Dynamic Graphs (Reviewer jFx8):**
The concern regarding the computational overhead of reconstructing physics-based edges for large-scale cities was addressed by a stress test. The authors showed that reconstructing edges for a 90,000-node region takes less than 40 seconds, proving the method's scalability.
4. **Methodological Justification (Reviewers mEEd, jFx8):**
Questions regarding the choice of GNNs over Transformers and the use of LSTMs were answered. The authors demonstrated that the "decoupled RGCN-LSTM" architecture outperforms attention-based baselines (ASTGCN, RGCN-Transformer) for this specific task, arguing that the recurrent structure better aligns with the Markovian nature of thermodynamic evolution.
5. **Rigidity of Physical Rules (Reviewer dxkG):**
The concern that hard-coded physical rules might limit flexibility was addressed by clarifying the role of learnable weights (Soft Correction) in the RGCN, which allow the model to adaptively suppress inaccurate heuristic connections.

Remaining Concerns:

1. **Lack of Final Confirmation:**
While Reviewer jFx8 explicitly confirmed that their concerns were addressed and maintained a positive score, Reviewers nd8D, dxkG, and mEEd did not respond to the rebuttal.
2. **Inherent Trade-off of Structure-based Bias:**
The reliance on predefined physical rules versus fully latent discovery (Reviewer dxkG) remains an inherent design choice, though the authors justified it for engineering reliability.

**Reviewer Scores:**

Reviewer jFx8, dxkG, and mEEd scored 6, and Reviewer nd8D scored 4. There are no clear indications that these reviewers will revise their final scores.

---

### Decision · Program_Chairs · 2026-01-26

Accept (Poster)